# Deep Tree Tensor Networks

**Chang Nie**
Nanjing University of Science and Technology
Nanjing, China
`changnie@njust.edu.cn`

## Abstract

Originating in quantum physics, tensor networks (TNs) have been widely adopted as exponential machines and parametric decomposers for recognition tasks. Typical TN models, such as Matrix Product States (MPS), have not yet achieved successful application in natural image recognition. When employed, they primarily serve to compress parameters within pre-existing networks, thereby losing their distinctive capability to capture exponential-order feature interactions. This paper introduces a novel architecture named *Deep Tree Tensor Network* (DTTN), which captures $2^L$-order multiplicative interactions across features through multilinear operations, while essentially unfolding into a *tree*-like TN topology with the parameter-sharing property. DTTN is stacked with multiple antisymmetric interaction modules (AIMs), and this design facilitates efficient implementation. Furthermore, our theoretical analysis demonstrates the equivalence between quantum-inspired TN models and polynomial/multilinear networks under specific conditions. We posit that the DTTN could catalyze more interpretable research within this field. The proposed model is evaluated across multiple benchmarks and domains, demonstrating superior performance compared to both peer methods and state-of-the-art architectures. Our code is publicly available at `https://github.com/NieCha/deep_tree_tensor_network`.

## 1 Introduction

*"Simplicity is the ultimate sophistication."* — Leonardo da Vinci

The wavefunction of a quantum many-body system typically resides in an extremely high-dimensional Hilbert space, with its complexity increasing exponentially as the particle count grows [27, 60]. For example, consider a system consisting of $N$ spin-$\frac{1}{2}$ particles; the dimensionality of the corresponding Hilbert space would be $2^N$. Tensor networks (TNs) offer powerful numerical techniques for tackling the "*Curse of Dimensionality*" [12]. By leveraging the local entanglement properties of quantum states, TNs represent complex wavefunctions into multilinear representations of multiple low-dimensional cores, thereby significantly reducing computational and storage requirements [11, 44][1]. This class of methods allows for an accurate representation of quantum states while mitigating the exponential growth in complexity, making it feasible to simulate large-scale quantum systems [26, 44].

Recently, TN-based interpretable and quantum-inspired white-box machine learning has attracted the attention of researchers. It holds the potential to generate novel schemes that can run on quantum hardware [25, 43]. Typical TN models, including Matrix Product States [13] (MPS[2]), Tree Tensor Network (TTN) [5], and Multi-Scale Entanglement Renormalization Ansatz (MERA) [45] are

---

[1]In tensor network theory, states that satisfy the area law for entanglement entropy can be efficiently approximated using TNs with finite bond dimensions.

[2]The MPS is also referred to as Tensor Train [41], or Tensor Ring [61] with the periodic condition in classical machine learning.

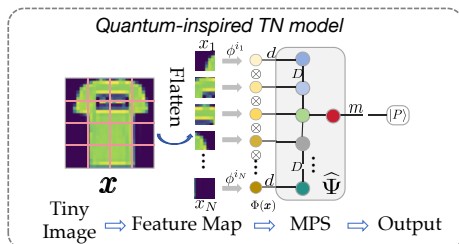
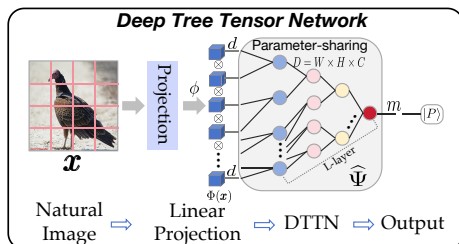

Figure 1: Schematic diagram of the quantum-inspired MPS model and DTTN towards image recognition task. The former is applied for tiny inputs and setting a small local mapping dimension $d = 2$ and bond dimension $D \leq 64$ in general[43]. DTTN handles complex inputs while retaining spatial locality in linear projection, and its parameter-sharing nature allows for maintaining a high bond dimension.

skillfully applied in image classification. Routine practice is to map each pixel or local patch of the image to a $d$-dimensional vector by a local map function $\phi(\cdot)$, and then use the *tensor product* to obtain a joint feature map $\Phi(\boldsymbol{x})$ of $d^N$ dimensionality (see Fig. 1 left). This process can be expressed as follows:

$$f(\boldsymbol{x}) = \arg\max_m \ \langle \boldsymbol{W}^m, \Phi(\boldsymbol{x}) \rangle. \tag{1}$$

Here, $\boldsymbol{W}^m$ represents a $(N + 1)$-th order tensor with an output index $m$; $f(\cdot) : \mathbb{R}^{w \times h \times c} \to \mathbb{R}$ denotes a multilinear function. In principle, mapping samples into exponential dimensional space to achieve linear separability instead of adopting activation functions is the essence of TNs [47, 43, 42]. However, existing methods are limited to simple tasks, e.g., MNIST and Fashion MNIST [5, 43], and we reveal that this is mainly due to 1) *low computational efficiency* and 2) *lack of feature self-interaction capability* (Section 3.3 for more details). Consequently, our goal is to address these challenges by applying TNs to complex benchmarks such as ImageNet-1K, thereby bridging the gap between TNs and modern architectures.

Alternatively, TNs are popularly employed to parameterize existing network models for acceleration. For example, the variational parameters of a network can be directly decomposed into TN formats, including convolutional kernels [40], fully-connected weights [32, 39], and attention blocks [34], to name a few. During inference, one can choose to retain or merge the TN structure as required. However, such techniques are devoid of probabilistic interpretation and feature multiplicative interaction. Notably, there exist intriguing and previously unexplored similarities between advanced deep learning architectures and TNs, such as MLP-Mixer [51], polynomial networks, and multilinear networks [9, 10, 7], which have demonstrated strong performance on complex visual tasks. As shown in Fig. 2, we visualize different architectures from the TN perspective for comparison. We note that contemporary architectures exhibit several key distinctions from existing TNs, including the incorporation of *nonlinear activations* and *instance normalization*. Consequently, we can improve TNs by drawing insights from advanced architectures through rigorous equivalence analysis, thereby overcoming their limitations on complex benchmarks.

Concretely, we introduce a novel class of TN architectures, named *Deep Tree Tensor Network* (DTTN), which uses multilinear operations to capture exponential-order interactions among features and predict targets without relying on activation functions or attention mechanisms. DTTN is constructed by sequentially stacking multiple antisymmetric interaction modules (AIMs) and inherently unfolds into a *tree*-like structure to reach on-par performance compared to advanced architectures, with better interpretability and understanding. Overall, the contributions of this paper are summarized as follows:

- We introduce DTTN, a simple yet effective architecture constructed by sequentially stacking AIMs. DTTN captures feature $2^L$ multiplicative interactions without activation functions and achieves state-of-the-art performance compared to other polynomial and multilinear networks with faster convergence (see Fig. 2 (e)).
- We provide a theoretical analysis of the equivalence between DTTN and other architectures under specific conditions. Without instance normalization, DTTN essentially reduces to a tree-topology TN, thereby overcoming the limitations of TN-based Born machines that excel mainly in simple tasks.

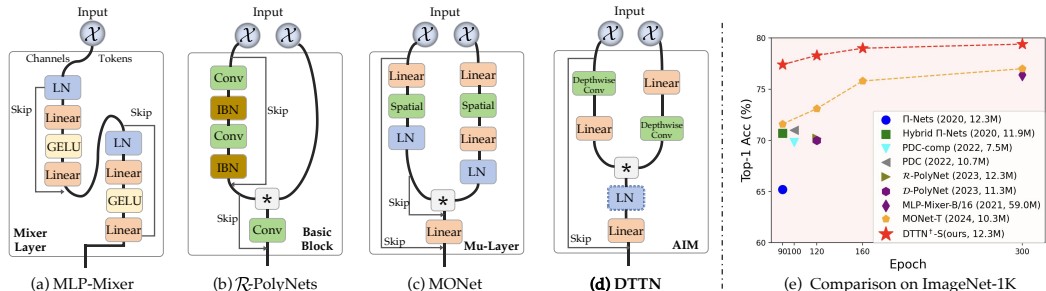

Figure 2: (a-d) Illustration of Core Blocks for Different Architectures. The MLP-Mixer utilizes GELU activation and other networks via the Hadamard product "∗" to enable the network to learn complex representations. We emphasize that instance batch normalization (IBN) and layer normalization (LN [57]) preceded before Hadamard product operations disrupt the polynomial unfolding nature of $\mathcal{R}$-PolyNets [10] and MONet [7]. In contrast, the succinctly designed AIM circumvents this issue. The optional LN inside AIM not only enhances performance but also facilitates faster convergence. (e) Comparison of Different Networks on ImageNet-1k. When comparing various networks trained on ImageNet over different epochs, DTTN stands out by achieving state-of-the-art performance compared to other multilinear networks, significantly outperforming them.

- We conduct a comprehensive evaluation of the proposed model's effectiveness and broader applicability across diverse benchmarks and domains, including visual recognition, recommender systems, and physics-informed neural networks. Our results show that DTTN achieves performance comparable to, and in certain cases even outperforms, carefully designed state-of-the-art architectures.

**Notations**. Throughout this paper, we use $\boldsymbol{x} \in \mathbb{R}^{I_1}$, $\boldsymbol{X} \in \mathbb{R}^{I_1 \times I_2}$, $\boldsymbol{\mathcal{X}} \in \mathbb{R}^{I_1 \times \cdots \times I_N}$ to denote first-order vectors, second-order matrices, and $N$-th order tensors, respectively. The blackboard letters are employed to represent a set of objects, e.g., $\mathbb{R}$ and $\mathbb{Z}$ denote real numbers and integers. In addition, $\ast$, $\odot$, and $\otimes$ denote the *Hadamard product*, *Khatri-Rao product*, and *tensor product*, respectively. For brevity, $|\mathbb{K}|$ denotes the cardinality of a set $\mathbb{K}$, and $\mathbb{K}_N$ denotes the positive integers set $\{1, 2, \ldots, N\}$. Moreover, for given tensors $\boldsymbol{\mathcal{A}} \in \mathbb{R}^{I_1 \times I_2 \times I_3}$ and $\boldsymbol{\mathcal{B}} \in \mathbb{R}^{I_4 \times I_5 \times I_6 \times I_7}$, with $I_2 = I_4$ and $I_3 = I_5$. The tensor contraction is executed by summing along the shared modes of $\boldsymbol{\mathcal{A}}$, $\boldsymbol{\mathcal{B}}$ to yield a new tensor $\boldsymbol{\mathcal{C}} = \boldsymbol{\mathcal{A}} \times_{2,3}^{1,2} \boldsymbol{\mathcal{B}} \in \mathbb{R}^{I_1 \times I_6 \times I_7}$. The entry-wise calculation can be expressed as $\boldsymbol{\mathcal{C}}_{(i_1, i_6, i_7)} = \sum_{i_2=1}^{I_2} \sum_{i_3=1}^{I_3} \boldsymbol{\mathcal{A}}_{(i_1, i_2, i_3)} \boldsymbol{\mathcal{B}}_{(i_2, i_3, i_6, i_7)}$. Refer to [29] for additional definitions.

## 2 Related Work

### 2.1 Quantum-Inspired TNs

Quantum-inspired tensor networks (TNs) enhance the performance of classical algorithms by mimicking quantum computational characteristics [25]. These methods map inputs into a Hilbert space with exponential dimensionality, achieving linear separability through local mappings and tensor products, while employing multiple low-order cores to parameterize coefficients, significantly reducing computational and storage complexity [47, 49]. The avoidance of activation functions, alongside the theoretical underpinnings rooted in many-body physics, contributes to the interpretability of TNs [43]. Recently, numerous studies have successfully applied TNs to tasks such as image classification [49], generation [5], and segmentation [48]. These studies effectively integrate established neural network techniques like residual connections [38], multiscale structures [35], and normalization [47] into TN frameworks. However, current TNs are predominantly suited for simpler tasks and face limitations in terms of computational efficiency and expressive power.

### 2.2 Advanced Modern Networks

In the contemporary landscape of deep learning, the design of network architectures has grown increasingly sophisticated and varied, each architecture presenting distinct advantages. Advanced models such as Convolutional Neural Networks (CNNs) renowned for efficient feature extraction [20,

22], Transformers distinguished by their powerful contextual understanding capabilities [54], MLP-based architectures celebrated for their simple yet effective designs [51], and Mamba noted for its linear complexity [18] have become pivotal across a wide array of applications. These networks leverage nonlinearities, while beneficial for model expressiveness, but limit applicability in domains such as security and encryption. Notably, Leveled Fully Homomorphic Encryption schemes can only support addition and multiplication operations [2, 7].

### 2.3 Polynomial and Multilinear Networks

Polynomial and multilinear networks employ addition and multiplication operations to construct intricate network representations [9, 10, 7]. Specifically, the pioneering Polynomial Network (PN) [9] constructs higher-order polynomial expansions of the input features in a modular fashion while supporting end-to-end training, achieving notable success in both image recognition and generation tasks. In their follow-up work, Chrysos et al. [10] introduce regularization strategies—such as data augmentation, instance normalization, and higher-order feature interactions—to further enhance model performance. Cheng et al. [7] drew inspiration from modern architectural designs to propose MONet, aiming to narrow the gap between multilinear networks and modern architectures. It is worth noting that both polynomial and multilinear networks can capture exponential-order feature interactions. However, a key distinction lies in their structural unfoldability: polynomial networks maintain an unfoldable structure, whereas multilinear networks may lose this property when LN is applied, as

$$\begin{cases} (\boldsymbol{A}\boldsymbol{z}) * (\boldsymbol{B}\boldsymbol{z}) = vec(\boldsymbol{z} \otimes \boldsymbol{z})(\boldsymbol{A}^T \odot \boldsymbol{B}^T), \\ \texttt{LN}(\boldsymbol{A}\boldsymbol{z}) * (\boldsymbol{B}\boldsymbol{z}) \neq vec(\boldsymbol{z} \otimes \boldsymbol{z}) \, \texttt{LN}(\boldsymbol{A}^T \odot \boldsymbol{B}^T). \end{cases} \tag{2}$$

Here $\texttt{LN}(\cdot)$ represents layer normalization, while $A$ and $B$ are learnable matrices.

In this work, we aim to achieve two objectives: (1) re-establish the polynomial expansion form and analyze the differences between multilinear and quantum-inspired TNs; and (2) develop a layer-normalized multilinear DTTN[†] that outperforms existing high-performance multilinear networks [7].

## 3 Method

In this section, we provide a detailed description of the Deep Tree Tensor Network (DTTN). We aim to construct a tree topology network by sequentially stacking AIM blocks, which consist solely of multilinear operations. For an input image $\boldsymbol{x}$, we apply a vanilla linear projection, also known as patch embedding [51], to obtain the feature map $\phi(\boldsymbol{x}, \boldsymbol{\Lambda}_\phi) \in \mathbb{R}^{W \times H \times C}$. Similar approaches are used in other methodologies. Here $\boldsymbol{\Lambda}_\phi \in \mathbb{R}^{S^2 \times C}$ represents a learnable matrix, where $S, C \in \mathbb{N}$ denote the local patch size and the number of output channels, respectively. This procedure corresponds to the local mapping illustrated in Fig. 1, with its output serving as the input for the DTTN. It should be noted that batch normalization (BN) operations following the linear layer have been omitted for brevity, as these operations can be integrated with the nearest linear layer during inference through structural re-parameterization technique [15].

### 3.1 Antisymmetric Interaction Module

The antisymmetric interaction module (AIM) is the core of DTTN. As illustrated in Fig. 2(d), for the $l$-th block input feature map $\boldsymbol{\mathcal{X}}^l \in \mathbb{R}^{W_l \times H_l \times C_l}$, we utilize an antisymmetric two-branch structure to capture the linear interactions of the input features separately. Both parts, denoted as $f_1^l, f_2^l$, incorporate a depthwise convolution layer and a linear layer, but apply them in reversed order. These layers are designed to capture spatial locality and channel interactions, respectively, effectively combining the advantages of CNN and MLP architectures [20, 51]. The antisymmetric design specifically targets reducing the complexity of AIM. The ratio of parameters and FLOPs between the two branches can be expressed as follows:

$$\begin{aligned} R_{Para} &= \frac{r_{exp}k^2 C_l + r_{exp}C_l^2}{r_{exp}k^2 C_l + r_{exp}^2 C_l^2} \quad \sim \quad \frac{1}{r_{exp}}, \\ R_{Flops} &= \frac{r_{exp}k^2 W_l H_l C_l + r_{exp}W_l H_l C_l^2}{r_{exp}k^2 W_l H_l C_l + r_{exp}^2 W_l H_l C_l^2} \quad \sim \quad \frac{1}{r_{exp}}, \end{aligned} \tag{3}$$

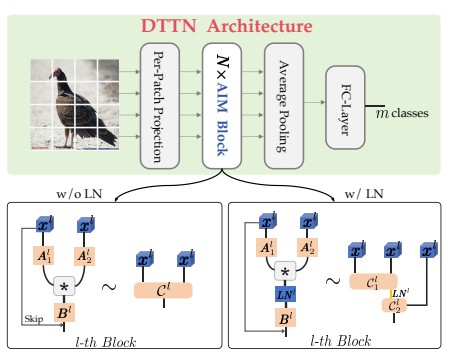

Figure 3: Schematic overview of the DTTN architecture.

Table 1: Specifications of different DTTN variants configuration. "Tiny", "Small", and "Large" refer to DTTN-T, DTTN-S, and DTTN-L configurations, respectively, each differing in parameter sizes. The primary distinction among these variants lies in the number of blocks and the hidden-sizes within their multi-stage structures. Additionally, models with layer normalization (LN) layers are denoted with a '†', such as DTTN†-S.

| Specification | Tiny | Small | Large |
|---|---|---|---|
| Numbers of Blocks | 34 | 44 | 56 |
| Hidden-size | 64,128,160,192 | 96,128,192,192 | 128,192,256,384 |
| Stages | 6,6,16,6 | 6,6,24,8 | 8,8,32,8 |
| Expansion ratio | 3 | 3 | 3 |
| Parameters (M) | 7.1 | 12.3 | 35.9 |
| FLOPs (B) | 2.3 | 4.1 | 12.3 |

where $r_{exp} \in \mathbb{N}_+$ represents the expansion ratio in the inverted bottleneck design, typically set to 3, $k \in \mathbb{N}_+$ denotes the kernel size, and $C_l, W_l, H_l$ are the number of channels, width and height of the $l$-th block input feature map, respectively. We use the hardware-friendly Hadamard product ($*$) to capture the second-order interactions of the branch outputs. Following this, an optional LN and a linear projection layer are sequentially applied to the computation results. Finally, a shortcut connection is used to preserve the input signal and accelerate training (see Fig. 2). Overall, the AIM forward calculation can be expressed as

$$\boldsymbol{\mathcal{X}}^{l+1} = \boldsymbol{\mathcal{X}}^l + Pro\left(f_1^l(\boldsymbol{\mathcal{X}}^l) * f_2^l(\boldsymbol{\mathcal{X}}^l)\right), \forall l \in \mathbb{K}_L, \tag{4}$$

where $Pro(\cdot)$ denotes the projection transformation operation. In summary, AIM captures second-order multiplicative interactions among input elements through multilinear operations without employing *nonlinear activations*. In contrast to core blocks inside other architectures, such as the Basic Block in $\mathcal{R}$-PolyNets [10] and Mu-layer[3] in MONet [7], AIM employs an antisymmetric design with only one shortcut connection.

### 3.2 Network Architecture

Our proposed architecture is constructed by stacking $L$ AIM blocks sequentially. As shown in Fig. 3, the final output is derived through average pooling and a fully-connected layer. The DTTN architecture is multi-stage, in line with its peers [9, 21, 7]. By varying the hidden-sizes and the number of blocks in each stage, we have designed three variants: *DTTN-T*, *DTTN-S*, and *DTTN-L* (see Table 1), each with distinct parameters to facilitate a comparative analysis. At a high level, we assert that the DTTN exhibits the following theoretical property:

**Proposition 1.** *The DTTN has the capability to capture $2^L$ multiplicative interactions among input elements, which can be represented in the format of Equation (1) as $\Phi(\boldsymbol{x}) = \otimes^{2^L} \phi(\boldsymbol{x}, \boldsymbol{\Lambda}_\phi)$. Consequently, the elements of $f(\boldsymbol{x})$ are homogeneous polynomials of degree $2^L$ over the feature map $\phi(\boldsymbol{x}, \boldsymbol{\Lambda}_\phi)$.*

It is important to note, however, that this characteristic no longer holds when LN is incorporated into the network, as LN introduces second-order statistical information. This phenomenon is clearly exemplified in models like MONet and DTTN†, where the network essentially behaves as a standard multilinear model.

**Unfolding Topology**. As illustrated in Fig. 3, the AIM is equivalent to a binary tree node in the absence of LN. In this case, AIM forward computation can be regarded as a TN contraction, which

---

[3]The MONet architecture comprises a stack of two variants of Mu-Layer. The second variant differs from the one shown in Fig. 2(c) in that it does not incorporate a spatial shift operation.

can be formulated as

$$
\begin{aligned}
\boldsymbol{x}^{l+1} &= \boldsymbol{x}^l + \boldsymbol{B}^l\left((\boldsymbol{A}_1^l \boldsymbol{x}^l) * (\boldsymbol{A}_2^l \boldsymbol{x}^l)\right) \\
&= \boldsymbol{x}^l + \texttt{Reshape}\left(\boldsymbol{B}^l(\boldsymbol{A}_1^{l\,T} \odot \boldsymbol{A}_2^{l\,T})^T\right) \times_{2,3}^{1,2} (\boldsymbol{x}^l \otimes \boldsymbol{x}^l) \\
&= \boldsymbol{\mathcal{C}}^l \times_{2,3}^{1,2} (\boldsymbol{x}^l \otimes \boldsymbol{x}^l)
\end{aligned}
\tag{5}
$$

Here $\boldsymbol{x}^l = vec(\boldsymbol{\mathcal{X}}^l) \in \mathbb{R}^{W_l H_l C_l}$, $\boldsymbol{\mathcal{C}}^l$ signifies a third-order tensor representing structural re-parameterization with the learnable matrices $\boldsymbol{A}_1^l$, $\boldsymbol{A}_2^l$ and $\boldsymbol{B}^l$. The AIM captures the second-order multiplicative interactions among input elements via *tensor product* and structures the three-order tensor utilizing *PyTorch* operators. Thus, one DTTN composed of $L$ AIMs can essentially be unfolded into a tree network with $2^L$ leaf nodes, as pictured in Fig. 1.

### 3.3 DTTN vs. Other Architectures

**DTTN & Polynomial Networks**. DTTN can be expressed in the same polynomial expansion form as Π-Net [9], which is given by

$$
f(\boldsymbol{x}) = \sum_{l=1}^{2^L}\left(\mathcal{W}^{[l]} \times_{2,\ldots,l+1}^{1,\ldots,l} (\otimes^l \phi(x))\right) + \beta,
\tag{6}
$$

where $\beta \in \mathbb{R}^m$ represents the constant term, and $\mathcal{W}^{[l]}$ is a $(l+1)$-th order learnable parameter tensor. Notably, equations (1) and (6) become equivalent upon introducing a bias term for the elements of the input vector $\boldsymbol{x}$. The networks exhibit different structured representations of the coefficients $\mathcal{W}^{[l]}$, for all $l \in \mathbb{K}_{2^L}$, due to their distinct network blocks and computational graphs.

**DTTN & Multilinear Networks**. We note that networks which exclusively involve multilinear operations—such as MONet, DTTN$^\dagger$, and $\mathcal{R}$-PolyNets—yet lack the polynomial expansion structure can be classified as multilinear networks.

**DTTN & Quantum-inspired TNs**. The main advantages of DTTN over Quantum-inspired TNs are twofold. (1) *The alternative of the local map function.* Existing TNs employ trigonometric functions for local mapping, which leads to the absence of higher-order ($\geq 2$) terms involving input elements in the network's unfolded form. This limitation results in a loss of feature self-interaction capabilities. (2) *Higher bond dimension induced by parameter-sharing properties.* Quantum-inspired TNs facilitate the parallel contraction of shared indices among $N$ cores within the same layer. However, due to limited memory capacity, the bond dimension must often be restricted to smaller values. The following theorem establishes the equivalence between DTTNs and TNs:

**Theorem 1.** *Given the local mapping function $\phi^{i_j}(x_j) = [x_j^0, \ldots, x_j^{2^L}]^T$, a polynomial network with the expansion form of Equation (6) can be transformed into a quantum-inspired TNs model with finite bond dimension.*

Since the internal cores of a TN can be decomposed into a "**core-diagonal factor-core**" structure via Higher-order Singular Value Decomposition (HOSVD) [29] and subsequently merged with connectivity cores, we regard the structural differences between DTTN and quantum-inspired TNs as negligible. We believe the above theorem not only establishes an equivalence between quantum-inspired TNs and modern architectures, but also offers insights for the future development of more interpretable and high-performance TN models.

## 4 Experiments

In this section, we provide a comprehensive assessment of the DTTN's effectiveness. Specifically, in Section 4.1, we perform experiments on a series of image classification benchmarks to validate the model's superiority over other multilinear and TN architectures. In Section 4.2, we show the broader impact of the DTTN across other domains, including recommendation system and partial differential equation (PDE) solving. Section 4.3 presents ablation studies aimed at examining the impact of various design choices. The paper concludes with a discussion of the model's limitations. Further analysis and detailed results are provided in the appendix.

Table 2: ImageNet-1K classification accuracy for various network architectures. Models that can be polynomially expanded are marked in red, whereas other multilinear models that do not include activation functions but incorporate layer normalization are marked in green. Our DTTN[†]-T outperforms the previous SOTA model MONet-T by 0.9% with fewer parameters and FLOPs.

| Model | Top-1(%) | Params (M) | FLOPs(B) | Epoch | Activation | Attention | Reso. |
|---|---|---|---|---|---|---|---|
| **CNN-based** | | | | | | | |
| ResNet-50 [20] | 77.2 | 25.0 | 4.1 | - | ReLU | ✗ | $224^2$ |
| $A^2$Net [4] | 77.0 | 33.4 | 31.3 | - | ReLU | ✓ | $224^2$ |
| AA-ResNet-152 [1] | 79.1 | 61.6 | 23.8 | 100 | ReLU | ✓ | $224^2$ |
| RepVGG-B2g4 [15] | 79.4 | 55.7 | 11.3 | 200 | ReLU | ✗ | $224^2$ |
| **Transformer- and Mamba-based** | | | | | | | |
| ViT-B/16 [16] | 77.9 | 86.0 | 55.0 | 300 | GeLU | ✓ | $224^2$ |
| DeiT-S/16 [53] | 81.2 | 24.0 | 5.0 | 300 | GeLU | ✓ | $224^2$ |
| Swin-T/16 [36] | 81.3 | 29.0 | 4.5 | 300 | GeLU | ✓ | $224^2$ |
| Vim-S [62] | 80.5 | 26.0 | - | 300 | SiLU | ✓ | $224^2$ |
| **MLP-based** | | | | | | | |
| MLP-Mixer-B/16 [51] | 76.4 | 59.0 | 11.6 | 300 | GeLU | ✗ | $224^2$ |
| MLP-Mixer-L/16 [51] | 71.8 | 507.0 | 44.6 | 300 | GeLU | ✗ | $224^2$ |
| CycleMLP-T [3] | 81.3 | 28.8 | 4.4 | 300 | GeLU | ✗ | $224^2$ |
| Hire-MLP-Tiny [19] | 79.8 | 18.0 | 2.1 | 300 | GeLU | ✗ | $224^2$ |
| ResMLP-24 [52] | 79.4 | 6.0 | 30.0 | 300 | GeLU | ✗ | $224^2$ |
| $S^2$MLP-Wide [59] | 80.0 | 71.0 | 14.0 | 300 | GeLU | ✗ | $224^2$ |
| $S^2$MLP-Deep [59] | 80.7 | 10.5 | 51.0 | 300 | GeLU | ✗ | $224^2$ |
| ViP-Small/14 [21] | 80.5 | 30.0 | 6.5 | 300 | GeLU | ✓ | $224^2$ |
| AFFNet [24] | 79.8 | 6.0 | 1.5 | 300 | ReLU | ✓ | $256^2$ |
| **Polynomial- and Multilinear-based** | | | | | | | |
| Π-Nets [9] | 65.2 | 12.3 | 1.9 | 90 | - | ✗ | $224^2$ |
| **DTTN-S(ours)** | **71.8 / 77.2** | 12.3 | 4.1 | 90 / 300 | - | ✗ | $224^2$ |
| Hybrid Π-Nets [9] | 70.7 | 11.9 | 1.9 | 90 | ReLU+Tanh | ✗ | $224^2$ |
| PDC [8] | 71.0 | 10.7 | 1.6 | 100 | ReLU+Tanh | ✗ | $224^2$ |
| PDC-comp [8] | 70.2 | 7.5 | 1.3 | 100 | ReLU+Tanh | ✗ | $224^2$ |
| $\mathcal{R}$-PolyNets [10] | 70.2 | 12.3 | 1.9 | 120 | - | ✗ | $224^2$ |
| $\mathcal{D}$-PolyNets [10] | 70.0 | 11.3 | 1.9 | 120 | - | ✗ | $224^2$ |
| MONet-T [7] | 77.0 | 10.3 | 2.8 | 300 | - | ✗ | $224^2$ |
| **DTTN[†]-T(ours)** | **77.9** | 7.1 | 2.3 | 300 | - | ✗ | $224^2$ |
| **DTTN[†]-S(ours)** | **79.4** | 12.3 | 4.1 | 300 | - | ✗ | $224^2$ |
| MONet-S [7] | 81.3 | 32.9 | 6.8 | 300 | - | ✗ | $224^2$ |
| **DTTN[†]-L(ours)** | **82.4** | 35.9 | 12.3 | 300 | - | ✗ | $224^2$ |

## 4.1 Visual Recognition

**Setup and Training Details**. A series of benchmarks with different types, scales, and resolutions are employed for the experiments, including CIFAR-10 [30], Tiny ImageNet [31], ImageNet-100 [58], ImageNet-1K [46], MNIST, and Fashion-MNIST [56]. A detailed description of the benchmarks and training configurations is included in the supplement.

Table 3: Experimental validation of various network architectures was conducted on smaller benchmarks with differing resolutions. The top-performing results are highlighted in bold. Among these, our DTTN[†]-S model achieves the best performance across all benchmarks, outperforming other polynomial and multilinear networks.

| | CIFAR-10 | Tiny ImageNet | ImageNet-100 |
|---|---|---|---|
| Resolution | $32^2$ | $64^2$ | $224^2$ |
| Resnet18 | 94.4 | 61.5 | 85.6 |
| MLP-Mixer | 90.6 | 45.6 | 84.3 |
| Res-MLP | 92.3 | 58.9 | 84.8 |
| Π-Nets | 90.7 | 50.2 | 81.4 |
| Hybrid Π-Nets | 94.4 | 61.1 | 85.9 |
| PDC | 90.9 | 45.2 | 82.8 |
| $\mathcal{D}$-PolyNets | 94.7 | 61.8 | 86.2 |
| MONet-T | 94.8 | 61.5 | 87.2 |
| DTTN[†]-S | **95.0** | **63.8** | **87.7** |

Table 4: Experimental validation was conducted comparing the DTTN-S with quantum-inspired TNs on the MNIST and Fashion-MNIST datasets. The highest performances, marked in bold, are achieved by our DTTN-S, which outperforms previous tensor networks, including aResMPS that utilizes residual connections and ReLU activation.

| Model | MNIST | Fashion-MNIST |
|---|---|---|
| MPS Machine | 0.9880 | 0.8970 |
| Bayesian TN | - | 0.8692 |
| PEPS | - | 0.883 |
| LoTeNet | 0.9822 | 0.8949 |
| aResMPS | 0.9907 | 0.9142 |
| DTTN-S | **0.9930** | **0.9236** |

**DTTN vs. Polynomial Networks**. Table 2 reports the results of the unfoldable networks DTTN-S and Π-Nets [9] on ImageNet-1K (light red areas), using neither activation functions nor instance

Table 5: Validating AIM as a pluggable module for enhancing feature interaction in recommendation models with consistent performance gains.

| Model | Criteo | Avazu |
|-------|--------|-------|
| DeepFM | 80.12 | 75.46 |
| DeepFM+AIM | $80.44_{+0.32}$ | $75.73_{+0.27}$ |
| FiBiNet | 80.42 | 76.01 |
| FiBiNet+AIM | $80.97_{+0.55}$ | $76.08_{+0.07}$ |
| DCN-V2 | 80.93 | 76.14 |
| DCN-V2+AIM | $81.15_{+0.22}$ | $76.52_{+0.38}$ |

Figure 4: Performance of PINNs on linear and nonlinear Allen-Cahn PDEs: L2 error and absolute error across the Spatial-Temporal domain.

Table 6: The influence of network depth and width on model performance.

| | Top-1 (%) | Params(M) |
|--|-----------|-----------|
| $L$=8, $d$=256 | 79.2 | 5.6 |
| $L$=16,$d$=256 | 85.5 | 10.2 |
| $L$=24,$d$=256 | 86.8 | 14.8 |
| $L$=32,$d$=256 | **87.2** | 19.4 |
| $L$=32,$d$=64 | 63.4 | 1.3 |
| $L$=32,$d$=128 | 82.5 | 4.9 |
| $L$=32,$d$=512 | **87.9** | 76.8 |

Table 7: The influence of different design choices for AIM on the performance of the DTTN variants.

| | Top-1 (%) | Params(M) |
|--|-----------|-----------|
| SIM-Conv | 86.2 | 9.1 |
| SIM-Linear | 84.9 | 4.8 |
| DTTN$^{\dagger}$-T | **86.4** | 6.9 |
| Sim-Conv | 87.8 | 15.9 |
| Sim-Linear | 85.2 | 8.3 |
| DTTN$^{\dagger}$-S | 87.7 | 12.1 |

Table 8: The influence of layer normalization inside AIM on the performance of the DTTN variants.

| Model | Top-1 (%) | Params(M) |
|-------|-----------|-----------|
| DTTN-T | 85.6 | 6.9 |
| DTTN$^{\dagger}$-T | $\mathbf{86.4}_{+1.8}$ | 6.9 |
| DTTN-S | 87.3 | 12.1 |
| DTTN$^{\dagger}$-S | $\mathbf{87.7}_{+0.4}$ | 12.1 |
| DTTN-L | 87.6 | 35.6 |
| DTTN$^{\dagger}$-L | $\mathbf{88.1}_{+0.5}$ | 35.6 |

normalization. DTTN-S achieves Top-1 accuracy of 71.8% and 77.2% after 90 and 300 training epochs, respectively, significantly outperforming Π-Nets by 5.6% and 12%, respectively. Additionally, DTTN-S maintains a significant advantage—nearly a 10% improvement in accuracy—over Hybrid Π-Nets that incorporate activation functions (which lose their unfolding properties).

**DTTN vs. Multilinear Networks**. Fig. 2(e) illustrates the performance of various multilinear networks trained on ImageNet-1K over different epochs, with the curve for DTTN$^{\dagger}$-S showing superior results compared to others. Tables 2 and 3 detail the Top-1 accuracies of multilinear architectures across a range of benchmarks of varying scales, including CIFAR-10, Tiny ImageNet, ImageNet-100, and ImageNet-1K. Specifically, DTTN-S achieves improvements of 0.2%, 2.3%, 0.5%, and 2.4% over the previous best models at similar scales. Moreover, the unfoldable DTTN-S attains an impressive 77.2% accuracy on ImageNet-1K without instance normalization. Additionally, DTTN$^{\dagger}$-T surpasses the prior state-of-the-art model MONet [7] by 0.9%, while utilizing approximately 30% fewer parameters and reducing FLOPs by 20%. These comparisons demonstrate that DTTN achieves significantly faster convergence and superior performance.

**DTTN vs. Quantum-inspired TNs**. DTTN outperforms other quantum-inspired TNs on smaller benchmarks like MNIST and Fashion-MNIST. Notably, these models have not yet been successfully scaled to large benchmarks. The baselines included for comparison are MPS [49], Bayesian TNS, PEPS [6], LoTeNet [47], and aResMPS [38]. The test results are reported in Tab. 4, where DTTN-S consistently achieves the best performance. Specifically, on the Fashion-MNIST dataset, DTTN-S outperformed the second-best model, aResMPS, by 0.96%. This superior performance can be attributed to the higher bond dimension facilitated by parameter sharing and self-interaction capabilities, as discussed in Section 3.3.

**DTTN vs. Advanced Architectures**. We further demonstrate the competitive performance of DTTN relative to modern architectures, marking it as the first TN model to be applied to large-scale benchmarks. Table 2 reports the Top-1 accuracy of the DTTN family compared to other models, including CNN-based, Transformer-based, Mamba-based, and MLP-based architectures

trained on ImageNet-1K. The DTTN$^\dagger$-L achieves an accuracy of 82.4% with 35.9M parameters, which is competitive and outperforms models such as DeiT-S/16 [53], ViP-Small/14 [21], and Vim-S [62]. Additionally, results in Tab. 3 further illustrate that DTTN achieves the highest accuracy on small benchmarks. The ViP-Small/14 [21] with an MLP architecture and MONet exhibit slower convergence on ImageNet-100 compared to ResNet-50 (see Appendix B), which is due to their inductive bias. In contrast, DTTN demonstrates remarkable training efficiency and superior accuracy relative to its counterparts.

## 4.2 Broader Impact

**Recommendation System**. Given the critical role of feature interaction in recommendation systems [33], we select two widely-used datasets, Criteo and Avazu, to evaluate the effectiveness of AIM in enhancing feature interaction modeling. In our implementation, AIM is designed as a pluggable module and seamlessly integrated into existing Click-Through Rate (CTR) prediction models, including DeepFM, FiBiNet [23], and DCN-V2 [55][4]. In this setup, AIM replaced all linear layers in the target models, with internal convolution operations removed, leaving only linear and normalization layers. Experimental validation, reported in Tab. 5 and evaluated using AUC, shows that AIM consistently enhances performance across both datasets for all tested CTR models. Specifically, FiBiNet saw a 0.55% improvement on the Criteo dataset.

**DTTN-based PDE Solver**. Physics-Informed Neural Networks (PINNs) [14] incorporate physical constraints of systems into the training process, enabling the model to learn governing physical laws for describing system behaviors. The nonlinear activation functions serve as the critical component for PINNs to capture complex patterns. Here, we apply DTTN to solve specific partial differential equations (PDEs), including both the linear case and the highly nonlinear Allen-Cahn equation (defined as $u_t = \epsilon\Delta u + u - u^3$, where $\epsilon$ controls interface width and $F(u) = u^3 - u$ represents nonlinear reaction terms. The two equations to be solved are as follows:

$$\begin{cases} \frac{\partial u}{\partial t} = -u \\ u(x, t = 0) = sin(\pi x) \\ u(1, t) = u(-1, t) = 0 \end{cases} \quad \begin{cases} u_t - 0.0001u_{xx} + 5u^3 - 5u = 0 \\ u(x, t = 0) = x^2 cos(\pi x) \\ u(1, t) = u(-1, t) = -1. \end{cases} \quad (7)$$

We employ PINN [14] and Self-Adaptive PINN (SA-PINN [50]) with hard boundary conditions to address these problems. For comparison, we replace the linear layers with the AIM module to construct PINN-DTTN and SA-PINN-DTTN, which maintain identical configurations to their baselines but omit the Tanh nonlinearity. Numerical results, including the average L2-error and the absolute error of the prediction in the spatial-temporal domain, are presented in Fig. 4. DTTN-based PINNs achieve lower prediction errors without activation nonlinearity.

## 4.3 Ablation Study

Here, we conduct ablation studies to evaluate the effectiveness of various design choices, including network depth and width, the antisymmetric design of AIM, and layer normalization. We utilize ImageNet-100—a representative subset of ImageNet-1K—as our test benchmark, which is well-suited for model validation and hyperparameter tuning under limited computational resources. Throughout these experiments, all hyperparameters are kept consistent with those of the final model, except for the variable under investigation. Each model was trained from scratch. We report the experimental results in Tables 6, 7, and 8.

**Depth and Width**. To control the depth and width of DTTN, we vary the number of AIM blocks and the hidden-size denoted as $L$ and $d$, for brevity. For ease of comparison, we kept $d$ fixed across different stages. As illustrated in Tab. 6, we first set $L$ to $\{8, 16, 24, 32\}$ with $d = 256$, and subsequently varied $d$ to $\{64, 128, 256, 512\}$ while fixing $L$ at 32. This approach enables us to explore how network depth and width affect model performance. Notably, performance declines significantly as both $L$ and $d$ are reduced. Specifically, Top-1 accuracy falls by 24.5% when $d$ is reduced from 512 to 64. Furthermore, the number of model parameters grows linearly with $L$ and quadratically with $d$. These observations provide indirect evidence for the effectiveness of DTTN's multi-stage design.

---

[4] `https://github.com/shenweichen/DeepCTR-Torch`

**Antisymmetrical Design of AIM**. We evaluate the effectiveness of the antisymmetric design of AIM on the DTTN$^\dagger$-T and DTTN$^\dagger$-S architectures. As shown in Tab. 7, we compare the impact of symmetric versus antisymmetric designs on model performance. In this context, SIM-Conv and SIM-Linear refer to the Symmetric Interaction Module (SIM), which uses prioritized convolutional and linear layers, respectively, as alternatives to AIM. We observe that DTTN$^\dagger$-T achieves the best results among its counterparts, whereas DTTN$^\dagger$-S performs only slightly worse than SIM-Conv (by 0.1%), while using approximately 20% fewer parameters. Furthermore, the reduction in the number of parameters aligns with the theoretical predictions corresponding to Equation 3.

**Importance of LN**. Figure 3(d) and Table 2 highlight the essential role of the Layer Normalization (LN) layer in enhancing the network's representational capacity—a key technique for improving the performance of multilinear networks [10, 7]. Table 8 reports the improvements in DTTN variants after incorporating LN, showing gains of 1.8%, 0.4%, and 0.5%, respectively. Moreover, by comparing the results in Fig. 2(e) and Tab. 2, we observe that DTTN-S achieves a Top-1 accuracy that is 5.6% and 2.2% lower than DTTN$^\dagger$-S after 90 and 300 training epochs, respectively.

Moreover, by combining the data from Figure 2(e) and Tab. 2, it can be seen that DTTN-S exhibits a Top-1 accuracy that is 5.6% and 2.2% lower than DTTN$^\dagger$-S after training for 90 and 300 epochs, respectively. This further underscores the importance of LN in boosting the performance and convergence of our proposed architecture.

**Limitations**. Although we have conducted an extensive evaluation of DTTN's effectiveness and advantages, computational resource limitations prevented us from performing additional experiments. These include pre-training on larger datasets like JFT-300M and evaluating model robustness. Future research will focus on a theoretical analysis of the proposed model, as well as investigate multilinear transformer architectures tailored for large language models (LLMs).

## 5 CONCLUSION

This paper introduces a multilinear network architecture named DTTN, which bridges the gap between quantum-inspired TNs and advanced architectures, achieving this uniquely without activation nonlinearity. Specifically, DTTN employs a modular stacking design to capture exponential interactions among input features, essentially unfolding into a tree-structured TN. We conducted extensive experiments to showcase the effectiveness of DTTNs across various benchmarks. Notably, this is the first validation of TNs' effectiveness on large-scale benchmarks, yielding competitive results compared to advanced architectures. Additionally, we explored the broader applicability of DTTN in other domains, such as recommendation systems and solving PDEs. Lastly, more theoretical discussions about DTTN will be addressed in future work.

## 6 Acknowledgment

We thank Yajie Chen and Junfang Chen for their valuable feedback and suggestions for improving this paper.

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

# Appendix

## A Experimental Setup

### A.1 Image Classification Benchmarks and Training Settings

A detailed description of the benchmarks and training configurations is provided below.

- CIFAR-10 [30] consists of 60K color images of $32 \times 32$ resolution across 10 classes, with 50K images for training and 10K for testing. We train our model using the SGD optimizer with a batch size of 128 for 160 epochs. The MultiStepLR strategy is applied to adjust the learning rate, and data augmentation settings are in accordance with [10].

- Tiny ImageNet [31], ImageNet-100 [58], and ImageNet-1K [46] contain 100k, 100k, and 1.2M color-annotated training images with resolutions of $64 \times 64$, $224 \times 224$, $224 \times 224$ pixels, respectively. These datasets serve as standard benchmarks for evaluating image recognition models. During training, we optimized our model using a configuration consistent with [7, 51][5]. Specifically, we utilized the AdamW optimizer [37] alongside a cosine decay schedule for learning rate tuning, and applied data augmentations including label smoothing, Cut-Mix, Mix-Up, and AutoAugment. Note that we did not employ additional large-scale datasets such as JFT-300M for pre-training, nor did we use unsupervised or semi-supervised methods to optimize our model. All experiments were conducted on 8 GeForce RTX 3090 GPUs using native PyTorch.

- MNIST and Fashion-MNIST [56] both contain 60,000 grayscale images for training and 10,000 for validation, with each image sized at $28 \times 28$. We conducted comparison experiments between DTTN and other TN models on these two benchmarks, employing training configurations consistent with those used for CIFAR-10.

### A.2 Hyperparameters for Training on ImageNet-1K

Table 11 shows the experimental hyperparameters used for training the DTTN family on the ImageNet-1K benchmark. We utilize the timm library[6] and ensure all settings are aligned with the comparison method MONet [7].

## B More Experimental Results

**Image Segmentation**. We employ the Semantic FPN framework [28][7] for performing semantic segmentation on the ADE20K dataset, which includes 20,000 training images and 2,000 validation images. DTTN$^{\dagger}$-S and DTTN$^{\dagger}$-L were initialized using pre-trained ImageNet-1K weights before being integrated as the backbone of the framework. Additionally, all newly added layer weights were initialized using Xavier initialization [17]. Table 9 presents the outcomes of the DTTN model trained for 12 epochs using the AdamW optimizer, evaluated by Mean Intersection over Union (mIoU). Some experimental data are derived from [7]. Notably, DTTN$^{\dagger}$-L achieves better performance over other multilinear networks, underscoring the efficacy of our designed AIM.

**Training Convergence**. Figure 5 displays the Top-1 accuracy and loss curves of the proposed model alongside other architectures—such as ResNet-50 [20], ViP-Small [21] representing the MLP architecture, and MONet [7]—all trained on ImageNet-100 for 90 epochs. It can be observed that DTTN achieves significantly faster convergence and superior performance.

## C Motivation and Interpretability

The principal motivation of this work is to uncover both the potential and inherent limitations of quantum-inspired tensor networks (TNs) on large-scale benchmarks, by establishing their equivalence with DTTN through Theorem 1. Although TNs have been extensively studied in the fields of quantum mechanics and white-box machine learning, state-of-the-art models are still largely restricted to small datasets and limited tasks. DTTN overcomes this barrier by extending Tensor Network Networks

---

[5]https://github.com/Allencheng97/Multilinear_Operator_Networks
[6]https://github.com/huggingface/pytorch-image-models
[7]https://github.com/CSAILVision/semantic-segmentation-pytorch

| BackBone | mIoU(%) |
|----------|---------|
| Resnet18 | 32.9 |
| FCN | 29.3 |
| Seg-Former | 37.4 |
| R-PDC | 20.7 |
| $\mathcal{R}$-PolyNets | 19.9 |
| MONet-S | 37.5 |
| DTTN$^\dagger$-S | 36.9 |
| DTTN$^\dagger$-L | **38.6** |

Table 9: Experimental validation of semantic segmentation on ADK20K with Semantic FPN.

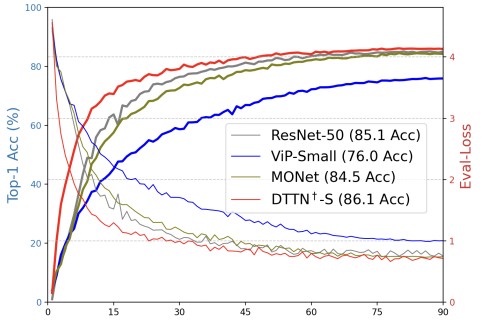

Figure 5: Top-1 accuracy and loss visualization for architectures trained from scratch on ImageNet-100.

(TNNs) to large-scale applications through techniques such as parameter sharing and exponential interaction modeling, thereby achieving competitive performance.

Furthermore, DTTNs retain a high degree of interpretability that aligns with quantum-inspired TNs, offering intuitive deterministic linear representations and probabilistic interpretations. For a deeper understanding of the theoretical underpinnings, readers are referred to related works [43, 44].

Table 10: Comparison of inference latency, throughput, and memory usage on an NVIDIA A6000 GPU.

| Model | Batch Size | Latency (ms/batch) | Throughput (sample/sec) | Peak Memory (GB) |
|-------|-----------|--------------------|-------------------------|------------------|
| MONet_T | 64 | 128.39 | 498.47 | 1.14 |
| ViP-Small/14 (ours) | 64 | 86.1 | 743.28 | 0.584 |
| DTTN-S (ours) | 64 | 81.94 | 781.08 | 1.099 |

## D  Latency Analysis

Table 10 presents a detailed comparison of inference latency and memory usage, benchmarked on an NVIDIA A6000 GPU. Compared to the previous state-of-the-art model, MONet-T, our DTTN-S reduces latency by nearly 35% while also achieving a 2.4% performance gain on ImageNet (see Table 1). Furthermore, DTTN demonstrates faster convergence on both the ImageNet and ImageNet-100 benchmarks, as shown by the convergence curves in Fig. 2e and Fig. 5.

Table 11: Training settings for ImageNet-1K in Section 4.1

| Item | Setting |
|------|---------|
| Optimizer | AdamW |
| Base learning rate | 1e-3 |
| Warmup-lr | 1e-6 |
| Learning rate schedule | cosine |
| Weight Decay | 0.01 & 0.02 |
| Batch size | $320 \times 4$ GPU |
| Label smoothing | 0.1 |
| Auto augmentation | ✓ |
| Random erase | 0.1 |
| Cutmix | 0.5 |
| Mixup | 0.5 |
| Dropout | 0.0 |

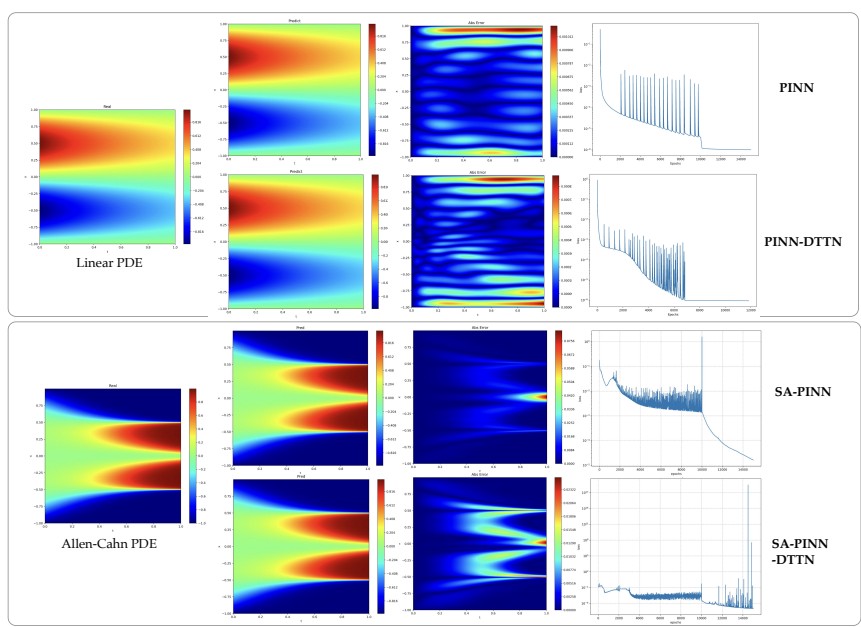

Figure 6: Visualization of training loss and prediction results of PINNs solving PDEs across the spatial-temporal domain.

## E   Details of DTTN Resolving PDEs

In this study, we use the Physics-Informed Neural Network (PINN) [14] and the Self-Adaptive PINN (SA-PINN) [50] as our baseline models. These are multi-layer feedforward networks with a hidden dimension of $d = 32$. Each linear layer is succeeded by a $\tanh$ activation function[8]. Subsequently, we replaced these linear layers with AIM modules while removing the activation functions, yielding the novel architectures PINN-DTTN and SA-PINN-DTTN. These enhanced networks are then applied to solve both the linear PDE and the nonlinear Allen-Cahn equation presented in Equation (8). As noted in reference [50], the Allen-Cahn PDE serves as an intriguing benchmark for testing PINNs, requiring the network to approximate solutions with sharp transitions in space and time, along with periodic boundary conditions.

Fig. 6 illustrates the prediction results and absolute errors for various PINN variants addressing the two equations across the spatial-temporal domain $\Omega \times T \to [-1, 1] \times [0, 1]$. For the first equation, which has an analytical solution of $u = e^{-t} \sin(\pi x)$, the PINN-DTTN exhibits lower errors in both training and prediction. However, for the second equation, despite the SA-PINN-DTTN achieving lower errors, there are noticeable spikes in training loss. This phenomenon may be due to instability when fitting complex functions with simple polynomials.

## F   Implementation and Motivation of the AIM Design

In Algorithm 1, we provide a PyTorch-style implementation of AIM. The main difference between training and inference lies in structural re-parameterization. Specifically, during training, BN layers are integrated with adjacent convolutional or linear layers. During inference, the AIM propagation process simplifies, matching the schematic shown in Fig. 2(d).

The primary motivation for designing DTTN was to develop a computationally efficient model, based solely on tensor operations, that is capable of scaling to large benchmarks. The core of our solution is the AIM, which is designed to facilitate direct multiplicative interactions. A tree structure emerges as a natural consequence of this design: since each AIM module performs a binary fusion of two feature tensors into one, stacking these modules hierarchically inherently constructs a tree. Significantly, this straightforward, hierarchical construction is provably equivalent to a powerful Tree Tensor Network

---

[8]https://github.com/ZzYyPp47/Attention_pinn

**Algorithm 1** Code for AIM (PyTorch-like)

```python
# H: height, W: width, C: channel, R: expansion ratio
# x: input tensor of shape (B, C, H, W)
############## Initialization ##############
proj_l = nn.Linear(C, C * R) # Aggregate channel information
conv_l = nn.Conv2d(C * R, C * R, kernel_size=3, groups=C * R) # Aggregate
        spatial information
proj_r = nn.Linear(C * R, C * R) # Aggregate channel information
conv_r = nn.Conv2d(C, C * R, kernel_size=3, groups=C) # Aggregate spatial
        information
proj = nn.Conv2d(C * R, C, kernel_size=1) # For information fusion
ln_norm = nn.LayerNorm([C * R]) # Layer normalization
l_norm = nn.BatchNorm2d(C * R) # Batch normalization
r_norm = nn.BatchNorm2d(C * R) # Batch normalization
res_norm = nn.BatchNorm2d(C) # Batch normalization
scale = nn.Parameter(torch.ones(1))
############## Training stage ##############
def AIM(x, use_ln):
    x_l = conv_l(proj_l(x.permute(0, 2, 3, 1).permute(0, 3, 1, 2)))
    x_r = proj_r(conv_r(x).permute(0, 2, 3, 1)).permute(0, 3, 1, 2)
    if use_ln:
        out = ln_norm(x_l * x_r)
    else:
        out = l_norm(x_l) * r_norm(x_r)
    out = res_norm(proj(out))
    return x + scale * out
############## Inference stage ##############
def AIM(x, use_ln):
    x_l = conv_l(proj_l(x.permute(0, 2, 3, 1).permute(0, 3, 1, 2)))
    x_r = proj_r(conv_r(x).permute(0, 2, 3, 1)).permute(0, 3, 1, 2)
    out = x_l * x_r
    if use_ln:
        out = ln_norm()
    return x + proj(out)
```

(TTN), bypassing the complex methods that have historically limited their application [5]. Thus, while the tree topology was not an initial design constraint, it is fundamental to our model's expressive power and theoretical grounding.

**Training Stability of DTTN.** The training of traditional TNs is often hindered by well-known challenges, such as gradient instability (i.e., exploding or vanishing gradients) and high sensitivity to hyperparameters. In contrast, our DTTN model trains stably across all scenarios, showing no signs of these issues, as evidenced by the smooth loss curves in Fig. 5. This stability provides a significant advantage over traditional TNs, where long chains of sequential tensor contractions can lead to optimization problems. Although DTTN is theoretically equivalent to a TTN, its hierarchical optimization process mirrors that of a standard deep network, ensuring a more robust and stable training dynamic.

## G  Complexity Analysis

We now analyze the complexity of DTTN in terms of its storage requirements and computational cost. Without loss of generality, we consider the multi-stage DTTN with hidden-size $d$ for each stage, and the total depth is $L$. We then consider the complexity of the main components of the network, including the local mapping, the core blocks, and the classification head separately.

In this section, we analyze the complexity of a DTTN, focusing on both storage requirements and computational cost. Without loss of generality, we consider a multi-stage DTTN where each stage has a hidden size of $d$, and the overall depth is $L$. We then examine the complexity associated with the main components of the network: the local mapping function, the core blocks, and the classification head.

**Local Mapping**. The local mapping function $\phi(\cdot)$ consists of two consecutive convolutional layers, each with a kernel size and stride of 2. This function is applied to the input image $\boldsymbol{x}$ to produce a feature map $\phi(\boldsymbol{x}, \boldsymbol{\Lambda_\phi}) \in \mathbb{R}^{W \times H \times C}$. Here, $\boldsymbol{\Lambda_\phi} \in \mathbb{R}^{S^2 \times C}$ represents a learnable matrix, where $S, C = d \in \mathbb{N}$ denote the local patch size and the number of output channels, respectively. The parameters and floating-point operations (FLOPs) involved in this process are given by:

$$
\begin{aligned}
Params^L &= \mathcal{O}(14 \cdot d + 4 \cdot d^2), \\
FLOPs^L &= \mathcal{O}(48 \cdot d \cdot WH + 4 \cdot d^2 \cdot WH).
\end{aligned}
\tag{8}
$$

**Core Blocks**. we analyze the complexity associated with the core AIM blocks within the DTTN architecture. Assuming that each of the four stages contains an equal number of blocks, the corresponding feature map sizes are $W \times H, \frac{W}{2} \times \frac{H}{2}, \frac{W}{4} \times \frac{H}{4}$, and $\frac{W}{8} \times \frac{H}{8}$, respectively. Our analysis focuses on the parameters and FLOPs involved in the three linear layers and two convolutional layers of AIM. The total parameters and FLOPs can be expressed as follows:

$$
\begin{aligned}
Params^{AIM} &= \mathcal{O}\left( (\sum_{s=0}^{3} 22 \cdot r_{exp} \cdot d + (2 \cdot r_{exp} + r_{exp}^2) \cdot d^2 + d) \cdot \frac{L}{4} \right) \\
&= \mathcal{O}\left( (22 \cdot r_{exp} + 2 \cdot (r_{exp} + r_{exp}^2) \cdot d + 1) \cdot d \cdot L \right), \\
FLOPs^{AIM} &= \mathcal{O}\left( (\sum_{s=0}^{3} 18 \cdot r_{exp} \cdot d \cdot \frac{WH}{4^s} + (2 \cdot r_{exp} + r_{exp}^2) \cdot d^2 \cdot \frac{WH}{4^s}) \cdot \frac{L}{4} \right) \\
&= \mathcal{O}\left( (\frac{765}{128} \cdot r_{exp} + \frac{85}{128} \cdot (r_{exp} + r_{exp}^2) \cdot d) \cdot d \cdot WH \cdot L \right).
\end{aligned}
\tag{9}
$$

**Classification Head**. For a classification head with $m$ classes, we receive a feature map of size $\frac{W}{8} \times \frac{H}{8} \times d$, which is then processed through an average pooling layer followed by a fully-connected layer to output an $m$-dimensional vector. The parameters and FLOPs involved in this process are given by:

$$
\begin{aligned}
Params^H &= \mathcal{O}(d \cdot (m+1)), \\
FLOPs^H &= \mathcal{O}(\frac{WH}{64} \cdot d + m \cdot d).
\end{aligned}
\tag{10}
$$

In summary, our conclusions regarding the number of parameters and FLOPs for the DTTN architecture can be expressed as:

$$
\begin{aligned}
Params &= Params^L + Params^{AIM} + Params^H, \\
FLOPs &= FLOPs^L + FLOPs^{AIM} + FLOPs^H.
\end{aligned}
\tag{11}
$$

It can be observed that the number of parameters in the DTTN architecture remains fixed, while the computational complexity exhibits a linear relationship with the input image scale. This characteristic represents a significant advantage of the DTTN over modern MLP and Transformer architectures, which typically exhibit quadratic complexity.

## H Proofs

Here, we derive the proofs of Proposition 1 and Theorem 1 from the main paper and further elucidate their significance.

**Proposition.** *The DTTN has the capability to capture $2^L$ multiplicative interactions among input elements, which can be represented in the format of Equation 1 as $\Phi(\boldsymbol{x}) = \otimes^{2^L} \phi(\boldsymbol{x}, \boldsymbol{\Lambda_\phi})$. Consequently, the elements of $f(\boldsymbol{x})$ are homogeneous polynomials of degree $2^L$ over the feature map $\phi(\boldsymbol{x}, \boldsymbol{\Lambda_\phi})$.*

*Proof.* For each AIM block with input $\boldsymbol{x}^l = vec(\boldsymbol{\mathcal{X}}^l) \in \mathbb{R}^{W_l H_l C_l}$, let $D^l = W_l \times H_l \times C_l$ and $l \in \mathbb{K}_L$. Suppose that the left and right branches of AIM can be represented as $f_1^l(\boldsymbol{x}^l) = \boldsymbol{A}_1^l \boldsymbol{x}^l$ and $f_2^l(\boldsymbol{x}^l) = \boldsymbol{A}_2^l \boldsymbol{x}^l$, where $\boldsymbol{A}_1^l, \boldsymbol{A}_2^l \in \mathbb{R}^{D^l \times D^l}$ are obtained through structured combinations of convolutional and linear layer weights. The feedforward propagation of the AIM block can then be

expressed as:

$$\begin{aligned}
\boldsymbol{x}^{l+1} &= \boldsymbol{x}^l + \boldsymbol{B}^l\left((\boldsymbol{A}_1^l\boldsymbol{x}^l) * (\boldsymbol{A}_2^l\boldsymbol{x}^l)\right) \\
&= \boldsymbol{x}^l + \texttt{Reshape}\left(\boldsymbol{B}^l(\boldsymbol{A}_1^{l\,T} \odot \boldsymbol{A}_2^{l\,T})^T\right) \times_{2,3}^{1,2} (\boldsymbol{x}^l \otimes \boldsymbol{x}^l) \\
&= \boldsymbol{x}^l + \boldsymbol{\mathcal{Z}}^l \times_{2,3}^{1,2} (\boldsymbol{x}^l \otimes \boldsymbol{x}^l).
\end{aligned} \tag{12}$$

where $\boldsymbol{B}^l$ denotes the fused linear layer inside AIM, and $\boldsymbol{\mathcal{Z}}^l = \texttt{Reshape}\left(\boldsymbol{B}^l(\boldsymbol{A}_1^{l\,T} \odot \boldsymbol{A}_2^{l\,T})^T\right) \in \mathbb{R}^{D^l \times D^l \times D^l}$ is a structured learnable tensor. Then each element of $\boldsymbol{x}^{l+1}$ can be calculated by

$$\boldsymbol{x}_\tau^{l+1} = \boldsymbol{x}_\tau^l + \sum_w^{D^l}\sum_\rho^{D^l} \boldsymbol{\mathcal{Z}}_{(w,\rho,\tau)}^l \boldsymbol{x}_w^l \boldsymbol{x}_\rho^l. \tag{13}$$

Thus, each AIM module captures second-order multiplicative feature interactions of the input. By induction, the DTTN stacked with $L$ AIMs captures $2^L$ interactions of the input $\boldsymbol{x}$. Note that the network's bias terms and shortcut connections can be eliminated by introducing an additional homogeneous dimension in the local mapping. Hence, we have $\boldsymbol{x}_\tau^{l+1} = \sum_w^{D^l+1}\sum_\rho^{D^l+1} \boldsymbol{\mathcal{Z}}_{(w,\rho,\tau)}^{*l}\boldsymbol{x}_w^l\boldsymbol{x}_\rho^l \in \mathbb{R}^{D_l+1}$. Therefore, the expression $f(\boldsymbol{x})$ is a homogeneous polynomial of degree $2^L$ of $\phi(\boldsymbol{x}, \boldsymbol{\Lambda}_\phi)$, which concludes our proof. $\square$

**Theorem.** *Given the local mapping function $\phi^{i_1}(x_1) = [x_1^0, \cdots, x_1^{2^L}]^T$, a polynomial network with the expansion form of Equation (6) can be transformed into a quantum-inspired TN model with finite bond dimension.*

*Proof.* For an input image $x \in \mathbb{R}^{W \times H \times C}$, the computation in the vanilla quantum-inspired TN model can be expressed as

$$f(\boldsymbol{x}) = TN\left(\{\phi^i(\boldsymbol{x}_{(\tau,\rho,w)})\}_{i=1}^N, \{\boldsymbol{\mathcal{R}}_i\}_{i=1}^N\right), \tag{14}$$

where $N = WHC, \tau \in \mathbb{K}_W, \rho \in \mathbb{K}_H, w \in \mathbb{K}_C$, and $\{\mathcal{R}_\rangle\}_{i=1}^N\}$ denotes the tensor cores. $TN(\cdot): \mathbb{R}^{\underbrace{(2^L+1)\cdots(2^L+1)}_{N}} \to \mathbb{R}^m$ represents the contraction operation that outputs an $m$-dimensional vector. We can further complete the contraction of $2^L$ physical indices and transform Equation 14 into

$$\begin{aligned}
f(\boldsymbol{x}) &= TN\left(\phi^1(\boldsymbol{x}_{(\tau,\rho,w)}) \times_1^1 \boldsymbol{\mathcal{R}}_1, \cdots, \phi^N(\boldsymbol{x}_{(\tau,\rho,w)}) \times_1^1 \boldsymbol{\mathcal{R}}_N\right) \\
&= TN(\boldsymbol{\mathcal{Z}}_1, \cdots, \boldsymbol{\mathcal{Z}}_N).
\end{aligned} \tag{15}$$

Since each output element of a polynomial network can be expressed as

$$\sum_{i_1}^N \cdots \sum_{i_{2L}}^N w_\xi x_{i_1} x_{i_2} \cdots x_{i_{2^L}}, \ \ \xi \le N^{2^L}. \tag{16}$$

The above equation is equivalent to

$$\sum_\xi w_\xi x_1^{k_1} \cdots x_N^{k_N} \quad where \ \ k_i \ge 0 \ \ and \ \ k_1 + \cdots + k_N = 2^L \tag{17}$$

Since Equation 15 encompasses each term of Equation 17, this proves our claim. $\square$

The above proof indicates that by selecting an appropriate local mapping, the DTTN can be transformed into a standard tensor network model. We re-emphasize that this is the first work demonstrating that tensor networks can achieve competitive performance on large-scale benchmarks, such as attaining 77.2% Top-1 accuracy on ImageNet-1K. Previous TNs have been limited to much smaller tasks due to expression limitations and lower bond dimensions. We hope that this research will inspire further explorations into tensor networks.

