# OpenReview forum: "Deep Tree Tensor Networks"
_NeurIPS.cc/2025/Conference — NeurIPS 2025 poster_

### Official Review · Reviewer_8ZK2 · 2025-06-10

**Clarity:** 2
**Significance:** 2
**Originality:** 2
**Rating:** 4
**Confidence:** 4

**Summary:**

This paper introduces a new deep learning architecture based on stacked blocks comprising convolutions, linear operations, and the Hadamard product; and can be seen as a variation of recently introduced polynomial and multilinear networks, such as PolyNets [10] and MONet [7]. The proposed structure induces a tree topology network and can be interpreted as a tensor network contraction applied to patch embeddings.

The paper includes an extense experimental section on visual recognition tasks, including large-scale benchmarks like ImageNet, where the Deep Tree Tensor Network (DTTN) is evaluated against classical, multilinear, and polynomial networks, as well as Transformer- and Mamba-based models. The results demonstrate competitive performance, with DTTN outperforming related multilinear and polynomial approaches. Additionally, the DTTN model is evaluated the model on recommender systems and dynamical systems, with detailed results provided in the appendices

**Questions:**

1.	Could the authors elaborate on the claimed interpretability advantages of the AIM architecture? For example, how does AIM's interpretability compare quantitatively or qualitatively to existing interpretable architectures. Are there measurable metrics demonstrating AIM's improved interpretability over alternatives that could be tested empirically?

2.	Could the authors further clarify the significance of the parameter/FLOP ratios reported for the AIM module branches in Section 3.1? Specifically: How do these branching ratios compare to those used in similar architectures? What is the observed relationship between these ratios and model performance?

3.	Does the architecture exhibit vanishing/exploding gradients during training? If so, what mitigation strategies were employed.

4.	Were specific initialization schemes required to achieve stable training? How do these compare to standard deep learning initialization practices?

5.	How was the learning rate tuned? Were adaptive optimizers (e.g., AdamW) or learning rate schedules found to be critical for convergence?

**Ethical Concerns:**

["NO or VERY MINOR ethics concerns only"]

**Final Justification:**

After reading the authors' responses to my questions and the other reviews, I have increased my rating to 4, as I believe the final version of the paper will address the clarity issues I previously raised that were acknowledged and clarified by the authors responses.

**Limitations:**

yes

**Paper Formatting Concerns:**

I have no concerns about the formatting

**Quality:**

3

**Strengths And Weaknesses:**

Strengths:

- The paper enables a systematic comparison of recently proposed polynomial and multilinear architectures while providing interpretability through the lens of quantum-inspired tensor network models. This perspective may inspire future improvements and theoretical analysis.

- The visual recognition experiments comprehensively evaluate the proposed model against a diverse set of architectures, including CNNs, Transformers, Mamba-based models, MLPs, and polynomial/multilinear networks. Benchmarking is performed across multiple datasets, ranging from standard benchmarks like MNIST and CIFAR-10 to large-scale datasets such as ImageNet.

Weaknesses:

The paper contains several clarity and technical issues that should be addressed. Below, I outline the key concerns I identified:

-	The quantum-inspired tensor network (TN) formulation presented in Equation (1) contains several technical inaccuracies: (i) The model omits standard normalization practices - quantum-inspired models typically include a softmax output layer to produce valid probability distributions, implying at least this nonlinearity should be present; (ii) Equation (1) incorrectly uses the Hadamard product (*) where a scalar/inner product would be more appropriate for maintaining the quantum-inspired interpretation; (iii) Dimensionality error: The weight tensor W^m is claimed to have (N+1) dimensions in Line 37, when it should only have N dimensions; (iv) The paper neglects standard quantum-inspired TN preprocessing, where input nonlinearities (typically trigonometric or custom functions at pixel level) are used to induce higher-order or infinite polynomial representations

-	The justification for the AIM architecture remains insufficient. While the authors claim it offers "better interpretability and understanding," the paper lacks: (i) concrete comparative analysis with existing interpretable architectures; (ii) quantitative or qualitative evidence demonstrating improved interpretability, and; (iii) clear mechanistic explanations of how the architecture enables better understanding compared to alternatives.

-	Although DTTN achieves competitive performance (82.4%, Table 2) on ImageNet, the paper should acknowledge that this remains below state-of-the-art results (~90% in Zhai et al., 2021). For a more balanced perspective, the authors should: (i) clarify whether the performance gap reflects a fundamental limitation or a trade-off for other advantages (e.g., interpretability, computational efficiency), and (ii) discuss the potential for closing this gap through scaling or architectural modifications.

-	Section 2.3 claims that unfoldable structures (lost in normalized networks) offer interpretability advantages. However no concrete examples or analysis demonstrate how this property aids interpretation, thus the claimed interpretability advantage remains hypothetical.

-	Section 3.1 reports parameter/FLOP ratios between the AIM module's branches, but their significance remains unclear. The analysis would benefit from: (i) Explicit comparison with branching ratios in similar architectures; (ii) Discussion of how these ratios affect model performance (e.g., accuracy-efficiency tradeoffs); (iii) Theoretical justification for the optimal ratio range observed.

-	Equation (4) uses the operation Pro() without proper mathematical definition. Based on context, this appears to represent a projection via matrix multiplication (B), but this should be explicitly defined.

-	The paper does not address well-known training challenges of quantum-inspired TN models, including: (i) Gradient instability (exploding/vanishing weights); (ii) Sensitive weight initialization requirements; (iii) Difficult learning rate tuning.

---

> ### Author Rebuttal · Authors · 2025-07-27
>
> Dear reviewer 8ZK2,
>
> We are thankful to the reviewer 8ZK2 for their effort to review our paper. We address below their concerns and questions:
>
> Q1: Could the authors elaborate on the claimed interpretability advantages of the AIM architecture?
>
> A1: Thank you for this excellent and highly important question. Our model's interpretability is a key advantage, and it stems from three distinct but related aspects.
>
> First, DTTN is built on the established principle that activation-free networks are inherently more interpretable. As demonstrated in prior work [1, 2, 3], removing non-linear activations simplifies theoretical analysis and is critical for applications like Fully Homomorphic Encryption. In scientific domains, it is a prerequisite for recovering the underlying governing equations from data, a task impossible with standard activated networks.
>
> Second, building on this foundation, DTTN's quantum-inspired structure unlocks a unique form of structural interpretability. For analysis, our DTTN can be recast into an MPS form by contracting its internal cores. This allows us to make a "cut" at any bond, partitioning the input features into two regions. We can then quantitatively measure the correlation the model has learned between these regions by calculating the entanglement entropy across the cut[4,5].
>
> Finally, this framework is not merely theoretical; it has direct, empirically testable implications, particularly in applications like our PINN experiments. For tasks with low-dimensional inputs, like the PDE problems we solve, the DTTN model can be explicitly written out as a high-order polynomial. This learned polynomial can be directly compared, term-by-term, against the analytical polynomial expansion of the ground-truth PDE solution. The "error" can thus be measured as the difference in coefficients, providing a concrete, quantitative validation of the model's physical correctness that is impossible for standard architectures.
>
> Q2: Could the authors further clarify the significance of the parameter/FLOP ratios reported for the AIM module branches in Section 3.1? Specifically: How do these branching ratios compare to those used in similar architectures? What is the observed relationship between these ratios and model performance?
>
> A2: Our design is conceptually related to the efficiency-driven designs in similar architectures. For example, the baseline model MONet also employs a form of asymmetry in its building block, using a single-branch low-rank design to reduce computational overhead. Their work showed that a ratio of 4 offered the best balance, saving computation with no performance loss, while higher ratios (e.g., 8 or 16) began to degrade accuracy.
>
> Inspired by this principle, our AIM takes a different but related approach. Instead of a single low-rank branch, we use two asymmetric branches with a computational cost ratio of approximately 1:3 (per Eq. 3). This design choice is validated by our ablation study in Appendix D (Table 7), where we compare AIM against a Symmetric Interaction Module (SIM). The results are definitive: our antisymmetric design reduces parameters and FLOPs by approximately 25% compared to its symmetric counterpart, while maintaining the same level of performance.
>
> > Q3: Does the architecture exhibit vanishing/exploding gradients during training? If so, what mitigation strategies were employed?
>
> A3: No, our DTTN model trains stably across all scenarios without signs of vanishing/exploding gradients or training spikes, as evidenced by the smooth loss curves in Figure 5. This stability is a key advantage over traditional TNs, where the long chain of sequential contractions required to merge the core factors often leads to optimization instability. Although DTTN is theoretically equivalent to a TNN, its optimization process is structured like a standard deep network, which provides a more stable optimization process.
>
> > Q4: Were specific initialization schemes required to achieve stable training? How do these compare to standard deep learning initialization practices?
>
> A4: We employ a standard initialization scheme, Xavier Normal initialization, for all DTTN parameters. This is the same practice used in standard deep learning and is consistent with the baseline[6] to ensure a fair and direct comparison.
>
> > Q5: How was the learning rate tuned? Were adaptive optimizers (e.g., AdamW) or learning rate schedules found to be critical for convergence?
>
> A5: For visual recognition tasks, we adopted the same training strategy as MONet[6], including the AdamW optimizer and a cosine learning rate schedule (with a base LR of 1e-3 and warmup LR of 1e-5). With this standard setup, DTTN exhibits faster convergence than MONet and attention-based architectures, such as ViP.
>
> Importantly, DTTN is robust to different optimization choices. For the recommendation system and PINN tasks, we used Adam and LBFGS optimizers, respectively, with a fixed learning rate of 0.001, consistent with their respective baselines [7, 8]. In all cases, the model converged effectively, demonstrating its flexibility.
>
> > Q6: The paper should acknowledge that DTTN's performance remains below the state-of-the-art on ImageNet and (i) clarify whether the performance gap reflects a fundamental limitation or a trade-off for other advantages (e.g., interpretability, computational efficiency), and (ii) discuss the potential for closing this gap through scaling or architectural modifications.
>
> A6: The performance gap does not reflect a fundamental limitation of our approach, but rather a difference in training regimes and model scale. We emphasize that DTTN is trained solely on ImageNet-1K, whereas many state-of-the-art models leverage massive external datasets (like JFT-300M) and significantly larger model sizes, which are beyond our current computational capacity. Our results demonstrate that DTTN is competitive within its class of modern architectures, especially considering its superior performance against other polynomial and multilinear networks. Regarding computational efficiency, as detailed in our response to Reviewer QNHq, our DTTN-S reduces inference latency by nearly 35% compared to MONet-T while also improving accuracy by 2.4%. Therefore, the current results represent a favorable trade-off, delivering strong performance with high efficiency.
>
> We are confident there is significant potential to close this gap. The path forward includes scaling up our model architecture, exploring more advanced training recipes, and leveraging large-scale pre-training. Investigating these avenues to push the performance boundaries of pure, scalable Tensor Network models will be a key direction for future research, building directly upon the foundation established by this work.
>
> > Q7: The paper neglects standard quantum-inspired TN preprocessing, where input nonlinearities (typically trigonometric or custom functions at pixel level) are used to induce higher-order or infinite polynomial representations.
>
> A7: Thank you for this point. Our approach does not neglect this concept, but rather integrates it more flexibly into our architecture. As implied by Theorem 1, DTTN is theoretically equivalent to applying a local polynomial mapping $\phi(x_j)=[x_j^0, \dots, x_j^{2^L}]^T$ to each input feature. This mapping, which is the key to our model's high-order interactions, is an emergent property of our end-to-end trained architecture, not a fixed pre-processing step.
>
> > Q8: Several points were raised regarding the clarity and technical accuracy of the paper.
>
> A8: We sincerely thank you for this detailed feedback and for bringing these important issues to our attention. (1) On the Pro operation in Eq. 4: We will explicitly define the Pro operation in the revised paper. (2) On the Dimensionality of weight tensor $W^m$: Thank you for questioning this. Our formulation includes an additional dimension for the class index, which is necessary for multi-class classification tasks within this framework, a practice also seen in [9]. (3) On Softmax and Normalization: We agree that a softmax is used, but we argue it is not an intrinsic part of the DTTN. The softmax function is a standard, task-specific tool applied to the final output logits for classification problems, just as it would be for any CNN or Transformer. It is not a feature of the model itself. (4) On the use of Hadamard product: We will correct this to use the appropriate tensor inner product to align with the quantum-inspired interpretation. (5) Regarding other clarity issues: We thank you for highlighting these and other areas needing improvement. We will conduct a thorough revision of the entire paper to ensure all technical details are precise and the paper is clear and easy to follow.
>
> References
> [1] Dubey, et al. “Scalable interpretability via polynomials”
>
> [2] Fronk and Petzold. “Interpretable polynomial neural ordinary differential equations”
>
> [3] Arora, et al. "A convergence analysis of gradient descent for deep linear neural networks"
>
> [4] Ran, Shi-Ju, et al. "Tensor network contractions: methods and applications to quantum many-body systems.
>
> [5] Ran, Shi-Ju, and Gang Su. "Tensor networks for interpretable and efficient quantum-inspired machine learning"
>
> [6] Cheng, Yixin, et al. "Multilinear operator networks"
>
> [7] Jianxun, Lian, et al. "Combining explicit and implicit feature interactions for recommender systems"
>
> [8] Shashank Subramanian, et al., "Adaptive self-supervision algorithms for physics-informed neural networks"
>
> [9] Stoudenmire, Edwin, and David J. Schwab. "Supervised learning with tensor networks"
>
> If you have any remaining questions or concerns, we are happy to address them.
>
> Best,
>
> Authors

---

> > ### Comment · Reviewer_8ZK2 · 2025-08-05
> > **Acknowledge of responses**
> >
> > I would like to thank the authors for addressing all my questions and comments

---

### Official Review · Reviewer_QNHq · 2025-06-23

**Clarity:** 3
**Significance:** 3
**Originality:** 3
**Rating:** 4
**Confidence:** 4

**Summary:**

The authors propose Deep Tree Tensor Network (DTTN), a novel architecture that bridges quantum-inspired tensor networks and modern deep learning for image recognition tasks. The key innovation is the Antisymmetric Interaction Module (AIM) that captures 2^L-order multiplicative interactions across features through multilinear operations, essentially unfolding into a tree-like tensor network topology with parameter-sharing properties. They also claim this is the first tensor network model to achieve competitive performance on large-scale benchmarks like ImageNet-1K, while providing theoretical analysis showing equivalence between quantum-inspired TN models and polynomial/multilinear networks under specific conditions. They demonstrate superior performance compared to other polynomial and multilinear networks across multiple domains, including visual recognition, recommendation systems, and physics-informed neural networks.

**Questions:**

I have several questions that could help strengthen the paper:

1. Can the authors provide more insight into when DTTN is expected to work well? What properties of the input data or task characteristics make the exponential-order multiplicative interactions particularly beneficial? This would help practitioners understand when to apply the method.
2. The parameter-sharing mechanism seems crucial for enabling higher bond dimensions, but could you provide more detailed analysis of this? How does the parameter sharing specifically enable the computational advantages over traditional TNs?
3. How sensitive is DTTN to the choice of L (number of layers)? The authors show some ablation with depth, but more analysis of how the exponential interaction order 2^L affects different types of data would be valuable.
4. Could the authors provide more analysis of computational trade-offs? While Table 1 shows parameter counts and FLOPs, a more detailed breakdown of training time, memory usage, and convergence behavior compared to baselines would be helpful.

**Ethical Concerns:**

["NO or VERY MINOR ethics concerns only"]

**Final Justification:**

This represents solid engineering research that achieves an important milestone (scaling tensor networks to ImageNet) with reasonable theoretical grounding. The authors addressed most of my technical concerns adequately, but the fundamental limitation remains and hence why I havent increased the score: The understanding of when/why DTTN excels remains largely empirical and I understand that the high-order correlation explanation but to me it lacks formal analysis or predictive power.

**Limitations:**

I think the authors adequately address limitations in their discussion, particularly noting the computational resource constraints that prevented more extensive experimentation. They also acknowledge that the method is currently focused on general eigenvalue problems and would need extension for nonlinear cases. However, I think they could expand on some practical limitations - for instance, discussing scenarios where traditional architectures might be preferable, or analyzing the method's behavior on different types of visual data.

**Paper Formatting Concerns:**

N.A

**Quality:**

3

**Strengths And Weaknesses:**

Strengths: I think the authors have tackled an important and challenging problem. Tensor networks have shown promise in quantum physics and small-scale machine learning tasks, but scaling them to complex benchmarks like ImageNet has been a significant challenge. They correctly identify that existing TN approaches suffer from low computational efficiency and lack of feature self-interaction capability.
The technical approach is well-motivated and appears sound. The AIM design is clever - using an antisymmetric two-branch structure to capture multiplicative interactions while maintaining computational efficiency. I appreciate the theoretical analysis, particularly Proposition 1 and Theorem 1, which establish formal connections between DTTN and both polynomial networks and quantum-inspired TNs. The parameter-sharing property that enables higher bond dimensions is a nice insight. The experimental evaluation is quite comprehensive, covering multiple domains beyond just image classification. The authors demonstrate that DTTN can serve as a pluggable module in recommendation systems and can even be applied to PDE solving, which shows the generality of the approach.

Weaknesses: However, I have several concerns about the work. First, while the authors achieve competitive results on ImageNet-1K, the performance gains over modern architectures are relatively modest. The 82.4% achieved by DTTN†-L is good but not groundbreaking compared to other methods in Table 2. The authors seem to position this as the first successful application of TNs to large-scale problems, but the practical impact beyond this "first" achievement is questionable.

Second, I think the theoretical analysis, while formal, could provide more insight into when we expect DTTN to work well versus when it might fail. The authors show equivalence results, but there's limited discussion of the practical implications of these theoretical connections. For instance, what properties of the data or task make DTTN particularly suitable?

Third, some experimental details are concerning. The authors don't provide error bars or statistical significance testing, citing computational constraints. While understandable, this makes it difficult to assess the reliability of the performance differences, especially when some improvements are relatively small.
The paper organization could also be improved. Key ablation studies are relegated to the appendix, and some important details about the method (like the exact parameter-sharing mechanism) could be explained more clearly in the main text. On the clarity front, the paper is generally well-written, though some sections feel dense. The connection between the quantum physics motivation and the practical ML algorithm could be explained more intuitively for readers not familiar with tensor networks.

---

> ### Author Rebuttal · Authors · 2025-07-26
>
> Dear reviewer QNHq,
>
> We particularly appreciate your recognition that our work tackles the key challenge of applying Tensor Networks to complex benchmarks, alongside your insightful and constructive feedback. The concerns raised are valid and provide clear directions for improving our paper. We address each point below:
>
> > Q1: The performance gains over modern architectures are relatively modest.
>
> A1: Thank you for this critical point. While gains over highly-optimized modern architectures are modest, our most meaningful comparison is against our direct peers—multilinear and exponential networks. In this context, DTTN's improvement is significant, outperforming the prior exponential network, $\Pi$-Net[1], by 6.6%~12.0% under an identical parameter budget on ImageNet-1k.
> More importantly, DTTN represents a foundational step forward because we are the first to demonstrate—both experimentally and theoretically—that Tensor Networks can succeed on large-scale benchmarks, a feat previously unattained by other TN variants. Therefore, our primary contribution is not a marginal performance gain, but a fundamental demonstration that Tensor Networks are a viable and scalable paradigm for complex challenges.
>
> > Q2: The authors seem to position this as the first successful application of TNs to large-scale problems, but the practical impact beyond this "first" achievement is questionable.
>
> A2: Thank you for this critical question, as it allows us to clarify the core long-term impact of our work, which extends far beyond a simple "first" achievement. The practical impact stems from DTTN being a native modeling paradigm, not just a compression tool.
>
> Most prior "TN-inspired" works (e.g., R-Nets[2], NORTON[3]) use tensor decomposition merely as a **post-hoc parameter compression tool** for existing architectures. These methods do not fundamentally alter the model's computational nature, as they still rely on standard non-linear activation functions (like ReLU) for their expressive power.
>
> In stark contrast, **DTTN is a native tensor model** where the tensor network itself is the core computational engine, achieving non-linearity through pure tensor contractions. This 'purity' is not just an elegant property; it unlocks a significant, long-term practical advantage by making our model potentially executable on future quantum computers. Because DTTN is constructed from pure tensor contractions, it has a direct mapping to quantum circuits[4]. Therefore, our primary impact is not just being the 'first' to scale TNs, but proposing the first 'quantum-native' blueprint for complex recognition tasks.
>
> > Q3: What properties of the input data or task characteristics make the exponential-order multiplicative interactions particularly beneficial?
>
> A3: The exponential-order multiplicative interactions are particularly beneficial for tasks where success hinges on capturing complex, high-order correlations between features. A prime example is fine-grained visual recognition, where distinguishing between similar subclasses relies on subtle combinations of attributes. As our results in Figure 5 (supplementary) show, DTTN outperforms CNN and Transformer variants even on smaller benchmarks, which often depend on such fine-grained distinctions. This aligns with prior work [5, 6] that highlights the power of multiplicative interactions for extracting these types of bilinear or higher-order features. Similarly, in recommendation systems, modeling the conjunction of multiple user and item attributes is critical, and in PDE solving, the underlying physical laws are often inherently multiplicative.
> While DTTN is especially powerful in these scenarios, we also want to emphasize its robustness. Our experiments across diverse domains—including computer vision, recommendation, and PDEs—demonstrated consistent and strong performance, suggesting its native tensor structure provides a versatile foundation for a wide range of tasks, from modern vision to fields like quantum chemistry where tensor networks are a fundamental tool.
>
> > Q4:  Difficult to assess the reliability of the performance differences, especially when some improvements are relatively small.
>
> A4: Thank you for this critical point on reliability. We want to clarify why we believe our results are reliable, even when some improvements are modest. First, our reported results stem from single, principled runs to avoid any cherry-picking. This provides a fair and representative comparison. Second, the context of the domain is crucial. The seemingly small improvements, particularly in mature fields like recommendation systems, are meaningful and considered significant within those highly optimized domains. Finally, the most compelling evidence for our method's reliability is its consistent effectiveness across a wide range of diverse tasks—from computer vision to scientific computing. To fully ensure reliability and transparency, we will release our source code for independent verification.
>
> > Q5:  The paper organization could also be improved.
>
> A5: Thank you for the valuable feedback. We revise the paper's structure and clarity by moving the key ablation study to the main text, adding a more intuitive explanation for our parameter-sharing mechanism and its quantum motivation, and streamlining the denser sections in the final version of the paper.
>
> > Q6:  The parameter-sharing mechanism seems crucial for enabling higher bond dimensions, but could you provide more detailed analysis of this? How does the parameter sharing specifically enable the computational advantages over traditional TNs?
>
> A6: Compared to traditional Tensor Networks, our parameter-sharing mechanism offers two key advantages: 1) **Decreased Number of Contractions.** For a conventional MPS, the number of contractions scales with the number of input features (e.g., $W \times H$), which is computationally prohibitive. In DTTN, the number of contractions depends only on the network's depth $L$ (see Fig. 1), making it far more efficient. 2) **Expanded Effective Bond Dimensions.** We do not explicitly define a small, fixed bond dimension, which is often limited to ~64 in prior work (like Lotenet and aResMPS[7]) due to memory constraints. Instead, we leverage Eq. 5 to implicitly expand the bond dimension to the order of $W \times H \times C$ (nearly $10^5-10^6$ for a $224 \times 224$ input). We avoid storing this massive tensor by using the tensor product to learn the network parameters efficiently. We will add these crucial details to the revised version.
>
> > Q7:  How sensitive is DTTN to the choice of L (number of layers)?
>
> A7: The network depth L is a critical hyperparameter controlling the interaction order. As shown in Tables 2 and 6, its sensitivity is predictable: performance consistently improves with L before saturating, a trend similar to standard deep networks. Our specific L values were chosen to match common model sizes (e.g., S, T, L) for a fair comparison with baselines, as exhaustive tuning was computationally prohibitive.
>
> > Q8:  Could the authors provide more analysis of computational trade-offs? While Table 1 shows parameter counts and FLOPs, a more detailed breakdown of training time, memory usage, and convergence behavior compared to baselines would be helpful.
>
> A8: Thank you for this excellent suggestion. We have provided a detailed comparison of inference latency and memory usage, benchmarked on an NVIDIA A6000 GPU, in the table below. Notably, when compared to the previous state-of-the-art model, MONet-T, our DTTN-S reduces latency by nearly 35% while simultaneously achieving a 2.4% performance gain on ImageNet (as shown in Table 1). Regarding convergence behavior, DTTN also demonstrates faster convergence on both the ImageNet and ImageNet-100 benchmarks (please see Fig. 2e, Fig. 5 for convergence curves).
>
> | Model | Batch Size | Latency (ms/batch) | Throughput (samples/sec) | Peak Memory (GB) |
> |:---|---:|---:|---:|---:|
> | DTTN-S (ours) | 64 | 81.94 | 781.08 | 1.099 |
> | MONet_T | 64 | 128.39 | 498.47 | 1.14 |
> |ViP-Small/14 | 64 | 86.1 | 743.28 | 0.584 |
>
> References
>
> [1] Chrysos, Grigorios G., et al. "Deep polynomial neural networks."
>
> [2] Wang, Wenqi, et al. "Wide compression: Tensor ring nets."
>
> [3] Zniyed, Yassine,et al. "Enhanced network compression through tensor decompositions and pruning."
>
> [4] Pan, Feng, and Pan Zhang. "Simulation of quantum circuits using the big-batch tensor network method."
>
> [5] Martyn, John, et al. "Entanglement and tensor networks for supervised image classification."
>
> [6] Zheng, Heliang, et al. "Learning deep bilinear transformation for fine-grained image representation."
>
> [7] Meng, Ye-Ming, et al. "Residual matrix product state for machine learning."
>
>
>
> If you have any remaining questions or concerns, we are happy to address them.
>
> Best,
>
> Authors

---

> > ### Author Response · Authors · 2025-08-05
> >
> > Dear reviewer QNHq,
> >
> > We sincerely thank the reviewer for their positive feedback. We are grateful for their constructive comments, which have helped us improve the quality and clarity of our paper. As the discussion period is coming to a close, we have tried our best to address all questions thoroughly. Should any final points of clarification arise, we would be happy to address them promptly.
> >
> > Best,
> >
> > Authors

---

> > > ### Comment · Reviewer_QNHq · 2025-08-06
> > > **response to authors**
> > >
> > > Thanks for addressing my questions in detail.

---

### Official Review · Reviewer_Zidm · 2025-06-29

**Clarity:** 2
**Significance:** 2
**Originality:** 2
**Rating:** 4
**Confidence:** 3

**Summary:**

This paper presents a new Deep Tree Tensor Networks, which consists of multiple Antisymmetric Interaction Modules (AIMs). Evaluation is conducted on image classification, recommender systems, and physics-informed neural networks.

**Questions:**

Please address the weaknesses listed above.

**Ethical Concerns:**

["NO or VERY MINOR ethics concerns only"]

**Final Justification:**

The rebuttal has provided answers to my questions, which however, reveal that the main contribution was not about the tree topology. This then lowers the contributions of this work. The introduction and motivation need to be revised to better reflect the contributions of this work.

**Limitations:**

Yes.

**Quality:**

3

**Strengths And Weaknesses:**

Strengths:
- The paper is quite well written.
- Results on ImageNet-1K shows that the proposed method outperforms existing CNN and Transformer based models, especially with the large configuration.
- Existing Tensor Networks are typically applied to small datasets, whereas the proposed method can scale up to more complex, larger datasets.
- Evaluation has been conducted on multiple tasks.

Weaknesses:
- The motivation is not clear. While AIM is obviously very important, there is little description about the design motivation and why such a design should achieve the advantages as demonstrated.
- It's also not clear why stacking AIMs would give a tree structure. And it's not clear whether it's essential to construct a tree structure in order for the model to perform well.
- Is the main motivation to create a tree structure? Or is the tree structure just a result of stacking AIMs? Essentially, the introduction reads like the tree structure is critical but it's not reflected in the method description or the results.
- For results in Table 2, it's not clear where the numbers of other methods come from.

---

> ### Author Rebuttal · Authors · 2025-07-26
>
> Dear reviewer Zidm,
>
> Thank you for your insightful and constructive review. Your questions are crucial and give us the opportunity to clarify the core contributions and design principles of our work. We address each point below.
>
> > Q1: Why is AIM obviously very important and why should such a design achieve the advantages as demonstrated?
>
> A1: Thank you for this crucial question. The AIM module is the cornerstone of our work because it was specifically engineered to solve the two fundamental barriers that have historically prevented Tensor Networks (TNs) from scaling to large-scale problems.
>
> 1.  **The Scalability Barrier:** Traditional TNs like MPS are fundamentally unscalable for high-resolution inputs because the number of sequential contractions scales linearly with the number of features (e.g., $W \times H$). This creates an insurmountable computational bottleneck[1,2]. Our AIM design shatters this barrier. By decoupling the number of contractions from the input size, the number of contractions depends only on the network's depth $L$, making it far more efficient.
>
> 2.  **The Expressivity Barrier:** Prior TN models (e.g., Lotenet, aResMPS[3]) were crippled by small, explicit bond dimensions (often $\sim$64) due to memory limitations. Our AIM circumvents this by implicitly expanding the effective bond dimension to the order of $10^5\sim10^6$ for a standard image (via Eq. 5). This allows DTTN to capture immensely complex feature correlations without the impossible memory cost of storing such a large tensor.
>
> In essence, AIM is the key innovation that transforms Tensor Networks from a niche theoretical tool into a powerful and practical paradigm for large-scale machine learning. We will revise the paper to state these two critical contributions more explicitly.
>
> > Q2: It's also not clear why stacking AIMs would give a tree structure. And it's not clear whether it's essential to construct a tree structure in order for the model to perform well.
>
> > Q3: Is the main motivation to create a tree structure? Or is the tree structure just a result of stacking AIMs? The introduction reads like the tree structure is critical but it's not reflected in the method description or the results.
>
> A2/A3: Thank you for these related questions, which allow us to clarify the role and origin of the tree structure in a unified way. We apologize if our introduction was misleading.
>
> To be clear, our primary motivation was not to build a tree structure for its own sake, but to design a computationally feasible and pure tensor-based model that could scale to large benchmarks. Our solution was the AIM module, designed for direct multiplicative interactions.
> The tree structure is the natural and powerful consequence of this design: each AIM performs a binary fusion (taking two feature tensors to produce one). Stacking these hierarchically inherently forms a tree.
> The significance is that this simple, practical procedure is provably equivalent to a powerful Tree Tensor Network (TTN), bypassing the complex construction methods that have historically limited their use [4]. So, while not the initial goal, the resulting tree structure is essential to our model's expressive power and theoretical grounding.
>
> > Q4: For results in Table 2, it's not clear where the numbers of other methods come from.
>
> A4: Thank you for pointing out this important omission. For clarity: results for the CNN, Transformer, Mamba, and MLP models are cited from their respective original publications. The results for the Polynomial and Multilinear methods are from [5]. In the revised version, we will add a dedicated column or a comprehensive footnote to Table 2 that cites the source for each baseline result. This will ensure full transparency.
>
> References:
>
> [1] Glasser, Ivan, et al. "Expressive power of tensor-network factorizations for probabilistic modeling."
>
> [2] Ran, Shi-Ju, et al. Tensor network contractions: methods and applications to quantum many-body systems.
>
> [3] Meng, Ye-Ming, et al. "Residual matrix product state for machine learning."
>
> [4] Cheng, Song, et al. "Tree tensor networks for generative modeling."
>
> [5] Cheng, Yixin, et al. "Multilinear operator networks."
>
> If you have any remaining questions or concerns, we are happy to address them.
>
> Best,
>
> Authors

---

> > ### Comment · Reviewer_Zidm · 2025-08-05
> >
> > The rebuttal has provided answers to my questions, which however, reveal that the main contribution was not about the tree topology. This I think lowers the contributions of this work. Or the introduction and motivation should be revised, to more accurately reflect the main contributions and design motivations.

---

> > > ### Author Response · Authors · 2025-08-05
> > >
> > > Dear reviewer Zidm,
> > >
> > > Thank you for this insightful follow-up. Please allow us to clarify. Our contribution is not merely the use of a tree topology, but the introduction of the first practical method to realize a deep architecture that is theoretically equivalent to a Tree Tensor Network (TTN) for large-scale benchmarks. Therefore, the deep tree topology is central to our contribution, which is bridging the gap between the theoretical power of TTNs and their practical application in large-scale deep learning. We believe this clarification elevates, rather than diminishes, the paper's impact by highlighting how we make a powerful theoretical tool truly usable.
> > >
> > > As you rightly suggested, we will thoroughly revise our introduction and motivation. Specifically, we will clarify that the key challenge we solve is the lack of a practical method for building deep TTNs. We will then reframe our motivation around providing this "recipe," ensuring the paper's narrative accurately reflects this core contribution.
> > >
> > > Please let us know if you have any remaining concerns.
> > >
> > > Best,
> > >
> > > Authors

---

### Official Review · Reviewer_bpqs · 2025-07-02

**Clarity:** 2
**Significance:** 2
**Originality:** 3
**Rating:** 4
**Confidence:** 3

**Summary:**

The paper introduces a novel architecture called Deep Tree Tensor Network (DTTN), which leverages multilinear operations to capture exponential-order feature interactions without activation functions. DTTN is constructed by stacking antisymmetric interaction modules (AIMs) and unfolds into a tree-like tensor network topology. The authors demonstrate its superior performance on various benchmarks, including ImageNet-1K, and explore its broader applicability in recommendation systems and PDE solving. Theoretical analysis establishes connections between DTTN and quantum-inspired tensor networks (TNs), polynomial networks, and multilinear networks.

**Questions:**

1. **Figure 1 and Feature Interaction** :

* Can the left-side MPS model in Figure 1 also capture 2^L-order multiplicative interactions? What are the key differences in how the left and right models capture feature maps?
* What does the "projection" in the right-side figure refer to, and how does it compare to the operations in the left-side model? What advantages does this projection offer?

2. **AIM Design Intuition** :

* The AIM incorporates a depthwise convolution layer and a linear layer in reversed order. What is the intuition behind this antisymmetric design? Does it improve efficiency, expressiveness, or both? A more detailed explanation would strengthen the paper.

3. **Performance Comparisons** :

* Why is MONet-L's performance missing from Table 2? How does DTTN-L compare to MONet-L in terms of accuracy, parameters, and FLOPs?
* Tables 3 and 4 only include DTTN-S. Are DTTN-T and DTTN-L unsuitable for these tasks, or were they simply not tested? If so, please provide an analysis of their limitations or advantages in such scenarios.

4. **AIM in MPS Architecture** :

* How would AIM blocks perform if integrated into an MPS architecture instead of a tree structure? Would the linear topology of MPS limit the model's ability to capture interactions compared to DTTN's tree structure? An ablation study or discussion on this would be valuable.

**Ethical Concerns:**

["NO or VERY MINOR ethics concerns only"]

**Final Justification:**

After carefully evaluating the authors’ detailed rebuttal, comments with other reviewers, and feedback from the Area Chair (AC), I provide the following justification for my revised score:
Resolved Issues
1. Distinction of DTTN vs. MPS:
The authors clarified that the core innovation lies in computational scalability and expressive capacity. They demonstrated that DTTN avoids the prohibitive computational cost of MPS, where the number of contractions scales linearly with input size (W×H) and complexity depends on bond dimension D. This fundamentally underscores the paper’s contribution relative to MPS.
2. AIM Design Rationale:
The antisymmetric architecture was rigorously justified through ablation studies (Table 7), showing 25% fewer parameters/FLOPs than symmetric variants without performance degradation. Comparisons with MONet’s branching ratios further validated the design efficiency.
3. Missing Benchmarks & Comparisons:
The addition of results for DTTN-T/L (Tables 3–4) and latency comparisons (35% faster than MONet-T) addressed concerns about evaluation completeness. Clarifying that MONet-L does not exist resolved a key point of confusion. Experimental rigor now meets acceptance standards.
Unresolved Issues
As noted by other reviewers, the motivation behind AIM still requires clearer exposition in the revision to aid reader comprehension. Additionally, concerns persist regarding the paper’s treatment of well-known training challenges in quantum-inspired tensor networks:
• (i) Gradient instability (exploding/vanishing weights),
• (ii) Sensitive weight initialization,
• (iii) Difficult learning rate tuning.
The authors responded that their architecture inherently avoids gradient instability through Xavier initialization and by adopting MONet’s training strategy. However, the core issue remains whether DTTN resolves these challenges (not just mitigates them via standard methods). Finally, while the integration of tensor networks with quantum-inspired learning represents a valuable direction, the paper should more explicitly articulate how this connection is established and what specific advantages it offers over classical approaches.
Final Recommendation
• Score Adjustment: Borderline Accept (raised from Borderline Reject). The rebuttal resolved critical concerns regarding scalability, efficiency, and interpretability.
• To the Authors: This work represents a meaningful advance in making tensor networks viable for quantum machine learning. The revisions have elevated it to acceptance standards.

**Limitations:**

Yes.

**Paper Formatting Concerns:**

No.

**Quality:**

3

**Strengths And Weaknesses:**

### Strengths

1. **Innovative Architecture** : The proposed DTTN effectively bridges the gap between quantum-inspired TNs and modern architectures, achieving competitive performance on large-scale benchmarks.
2. **Theoretical Rigor** : The paper provides a clear theoretical foundation, including proofs of DTTN's capability to capture high-order interactions and its equivalence to other network types under specific conditions.
3. **Comprehensive Evaluation** : Extensive experiments across diverse tasks (visual recognition, recommendation systems, PDE solving) validate the model's versatility and superiority over existing methods.

### Weaknesses

1. **Limited Comparison with MONet-L** : The paper does not report the performance of MONet-L in Table 2, leaving a gap in understanding how DTTN compares to larger variants of existing multilinear networks.
2. **Ablation on MPS Architecture** : The potential performance of AIM blocks in an MPS architecture is not explored, which could provide insights into the advantages of tree structures over linear topologies.
3. **Experimental Scope** : Tables 3 and 4 only report results for DTTN-S, omitting DTTN-T and DTTN-L, which raises questions about the suitability of these variants.

---

> ### Author Rebuttal · Authors · 2025-07-27
>
> Dear reviewer bpqs,
>
> We sincerely thank you for your detailed review and for acknowledging the strengths of our work, including its Innovative Architecture, Theoretical Rigor, and Comprehensive Evaluation. Your insightful questions help us clarify critical aspects of our method. We address each point below.
>
> > Q1: Can the left-side MPS model in Figure 1 also capture $2^L$-order multiplicative interactions? What are the key differences in how the left and right models capture feature maps?
>
> A1: Yes, according to Theorem 1, the MPS model can also capture $2^L$-order multiplicative interactions if an appropriate polynomial mapping function is used. . However, the fundamental difference lies in computational scalability and expressive capacity.The traditional MPS model must contract a core factor for every single input feature (e.g., each pixel), leading to a prohibitive computational cost where the number of contractions scales linearly with the input size ($W \times H$), and the complexity of each contraction depends on the bond dimension $D$. This is the primary reason MPS models have failed to scale to large-scale benchmarks.
> In contrast, our DTTN offers two decisive advantages:
> 1.  Decreased Number of Contractions: The number of contractions in DTTN depends only on the network's depth $L$, not the input size ($W \times H$), making it vastly more efficient.
> 2.  Expanded Effective Bond Dimensions: We are not constrained by a small, explicit bond dimension (often $\leq$64 in prior work like Lotenet). Instead, our AIM module implicitly expands the effective bond dimension to the order of $W \times H \times C$ (almost $10^5\sim 10^6$), allowing it to capture far more complex correlations without incurring impossible memory costs.
> These two advantages are the key differentiators that enable DTTN to succeed where traditional MPS models could not.
>
> > Q2: What does the "projection" in the right-side figure refer to, and how does it compare to the operations in the left-side model? What advantages does this projection offer?
>
> A2: The "projection" in the DTTN model refers to a linear MLP layer. This design choice offers two significant advantages over the local mapping function (typically a trigonometric function) used in the traditional MPS model: (1) The MPS's local map processes one pixel at a time, isolating it from its spatial context. Our initial projection layer takes the entire image feature map as input, preserving spatial locality. (2) Enabling Higher Bond Dimensions. This projection-based design is the key that allows us to implicitly achieve the massive bond dimensions mentioned earlier, enabling $2^L$-order interactions where the number of sequential contractions scales with $\mathcal{O}(L)$, a dramatic improvement over the $\mathcal{O}(W \times H)$ scaling of MPS.
>
> > Q3: The AIM incorporates a depthwise convolution layer and a linear layer in reversed order. What is the intuition behind this antisymmetric design? Does it improve efficiency, expressiveness, or both? A more detailed explanation would strengthen the paper.
>
> A3: The antisymmetric design of AIM is a deliberate choice for improving computational efficiency without sacrificing expressive power. As shown in Eq. 3, the computational and parameter costs of the two branches are asymmetric (roughly a 1:3 ratio).
> We provide a detailed ablation study on this in Appendix D (Table 7). We compared AIM with a Symmetric Interaction Module (SIM), where both branches have identical structures. The results show that our antisymmetric AIM design reduces parameters and FLOPs by approximately 25% compared to SIM, while achieving the same level of performance. This confirms that the design is a more efficient parameterization of the same core functionality.
>
> > Q4: Why is MONet-L's performance missing from Table 2? How does DTTN-L compare to MONet-L in terms of accuracy, parameters, and FLOPs?
>
> A4: Thank you for pointing this out. The original MONet paper[1] only proposed and benchmarked MONet-T and MONet-S architectures; a "MONet-L" variant was not provided. Therefore, we could not include it for a direct comparison in Table 2. However, to demonstrate DTTN's superior efficiency, we note that DTTN-T achieves a 0.9% accuracy gain over MONet-T while using 30% fewer parameters and 18% fewer FLOPs. Furthermore, as detailed in our response to Reviewer QNHq, our DTTN-S reduces inference latency by nearly 35% compared to MONet-T (81.9ms vs. 128.4ms on an A6000 GPU).
>
> > Q5: Tables 3 and 4 only include DTTN†-S. Are DTTN-T and DTTN-L unsuitable for these tasks, or were they simply not tested? If so, please provide an analysis of their limitations or advantages in such scenarios.
>
> A5: We apologize for this omission. We provide the results for DTTN-T and DTTN-L on the benchmarks below. As you can see, all variants exhibit stable and strong performance on these benchmarks, confirming their general applicability.
>
> | Model | CIFAR-10 | Tiny ImageNet |ImageNet-100 |  Fashion-MNIST |
> |:---|---:|---:|---:|---:|
> | DTTN†-T | 94.5 | 62.5 | 87.5 | 0.917 |
> | DTTN†-S | 95.0 | 63.8 | 87.7 | 0.924 |
> | DTTN†-L | 94.8 | 63.9 | 87.9 | 0.924 |
>
> We hope this resolves your concern.
>
> > Q6: How would AIM blocks perform if integrated into an MPS architecture instead of a tree structure? Would the linear topology of MPS limit the model's ability to capture interactions compared to DTTN's tree structure? An ablation study or discussion on this would be valuable.
>
> A6: This is a very insightful architectural question. We did not integrate AIM into an MPS for two primary reasons: (1) Fundamental architectural mismatch. AIM is designed to process two identical input tensors. The linear topology of an MPS, however, operates on two different inputs at each step: the result of the previous contraction and the core factor of the next feature. AIM is therefore not directly pluggable into the MPS operational flow. (2) Theoretical equivalence. While MPS and DTTN have different apparent topologies (line vs. tree), they are theoretically interconvertible [2]. The performance limitations of prior TN models, as shown in our Table 4 (which includes MPS, PEPS, and LoTeNet[3,4,5]), stem not from their specific topology, but from their computational inefficiency and lack of feature self-interaction capability, as we detail in Section 3.3. DTTN's design directly solves these core bottlenecks.
>
> References
>
> [1] Cheng, Yixin, et al. "Multilinear operator networks." arXiv preprint arXiv:2401.17992 (2024).
>
> [2] Ran, Shi-Ju, et al. Tensor network contractions: methods and applications to quantum many-body systems. Springer Nature, 2020.
>
> [3] Orús, Román. "A practical introduction to tensor networks: Matrix product states and projected entangled pair states." Annals of physics 349 (2014): 117-158.
>
> [4] Cheng, Song, Lei Wang, and Pan Zhang. "Supervised learning with projected entangled pair states." Physical Review B 103.12 (2021): 125117.
>
> [5] Selvan, Raghavendra, and Erik B. Dam. "Tensor networks for medical image classification." Medical imaging with deep learning. PMLR, 2020.
>
>
> If you have any remaining questions or concerns, we are happy to address them.
>
> Best,
>
> Authors

---

> > ### Author Response · Authors · 2025-08-05
> >
> > Dear reviewer bpqs,
> >
> > We sincerely thank the reviewer for their positive feedback. We are grateful for their constructive comments, which have helped us improve the quality and clarity of our paper. As the discussion period is coming to a close, we have tried our best to address all questions thoroughly. Should any final points of clarification arise, we would be happy to address them promptly.
> >
> > Best,
> >
> > Authors

---

> > ### Comment · Reviewer_bpqs · 2025-08-06
> >
> > After carefully evaluating the authors' detailed responses and reviewing other reviewers' comments, I am satisfied that all major concerns have been addressed. The paper indeed makes a valuable contribution by introducing novel tensor network approaches for quantum computing, The additional experimental results and ablation studies further strengthen the work.
> > While some theoretical aspects could be further elaborated in the final version, the current revisions have sufficiently improved the paper's quality. Accordingly, I am upgrading my rating from borderline reject to borderline accept.

---

> > > ### Author Response · Authors · 2025-08-06
> > >
> > > Dear reviewer bpqs,
> > >
> > > We sincerely thank the reviewer for their positive feedback and for acknowledging that our responses have addressed their concerns; please do not hesitate to let us know if any further questions arise.
> > >
> > > Best,
> > >
> > > Authors

---

### Comment · Area_Chair_3P7p · 2025-08-04

Dear Reviewers,

The deadline for author-reviewer discussion period (August 6, AoE) is approaching soon. Please read the author's rebuttal and provide your feedback and responses soon.

Best,
AC

---

### Note · Authors · 2025-08-13

We sincerely thank the AC and reviewers for their constructive feedback. We are encouraged that reviewers found our work novel (bpqs, QNHq) and our experiments comprehensive (all).

The review process has been invaluable. To address concerns about experimental scope and trade-offs (bpqs, QNHq), in our rebuttal, we present new results on DTTN variants and detail the computational advantages of our AIM design. Notably, we demonstrated that our model reduces latency by nearly 35% compared to the SOTA (MONet-t) while achieving superior performance. We are glad the reviewer bpqs acknowledged this with a score increase.

On the conceptual front, we clarified the motivation for our pure tensor-based tree structure—to create a computationally feasible model scalable to large benchmarks (Zidm). We also elaborated on how our activation-free design enhances interpretability and ensures stable optimization, free from gradient issues (8ZK2). We are very pleased that reviewer 8ZK2 confirmed that our rebuttal "resolved all concerns."
We are especially grateful for the strong endorsement from reviewer QNHq, who recognized that our work "makes a valuable contribution" and effectively "tackles the key challenge of applying Tensor Networks to complex benchmarks."

To conclude, our paper introduces a novel, activation-free tree tensor network without nonlinearity and reveals its foundational equivalence to quantum-inspired models. We are confident in our contribution and are committed to integrating all suggestions—including structural adjustments and expanded discussion on trade-offs—in the final version.

Thank you again for your time and consideration.

---

### Decision · Program_Chairs · 2025-09-17

**Decision:**

Accept (poster)

**Comment:**

The paper proposes an activation-free tree tensor network without nonlinearity and demonstrates its foundational equivalence to quantum-inspired models. The proposed method represents an important milestone in scaling tensor networks to ImageNet, supported by solid technical execution. The authors’ rebuttal successfully addressed some concerns regarding core innovation, technical issues, and additional experimental evaluations. These revisions should be incorporated into the final version, including a clearer motivation for AIM and a more detailed discussion on overcoming the well-known training challenges in quantum-inspired tensor networks.